# A SET-SEQUENCE MODEL FOR TIME SERIES

## ABSTRACT

Many prediction problems across science and engineering, especially in finance and economics, involve large cross-sections of individual time series, where each *unit* (e.g., a loan, stock, or customer) is driven by unit-level features and latent cross-sectional dynamics. While sequence models have advanced per-unit temporal prediction, capturing cross-sectional effects often still relies on hand-crafted summary features. We propose *Set-Sequence*, a model that *learns* cross-sectional structure directly, enhancing expressivity and eliminating manual feature engineering. At each time step, a permutation-invariant Set module summarizes the unit set; a Sequence module then models each unit's dynamics conditioned on both its features *and* the learned summary. The architecture accommodates unaligned series, supports varying numbers of units at inference, integrates with standard sequence backbones (e.g., Transformers), and scales linearly in cross-sectional size. Across a synthetic contagion task and two large-scale real-world applications—equity portfolio optimization and loan risk prediction—Set-Sequence significantly outperforms strong baselines, delivering higher Sharpe ratios, improved AUCs, and interpretable cross-sectional summaries.

## 1 INTRODUCTION

Many key problems in finance—such as constructing stock portfolios or estimating risk for a pool of loans—involve predicting outcomes for large populations of correlated units. Similar problems appear in science and engineering where the individual units may be sensors, devices, or customers. A key characteristic of such problems is they involve a varying number of correlated units, with each unit represented by a time series of dynamic features. In short, they involve a *set of sequences* where the sequences share the time axis. This poses a two-dimensional challenge: capturing dependencies across the *cross-sectional dimension* of $M$ units, while modeling the *temporal dynamics* over $T$ time steps. Directly modeling the joint distribution of the $M$ units is intractable due to its size. For example, a portfolio of $M = 10,000$ loans where each loan has $50$ features per time step, results in a (joint) time series with $50,000$ features per time step (Anenberg & Kung (2014); Azizpour et al. (2018)). To avoid this, existing approaches make predictions for each unit separately, while augmenting the unit's own features with hand-crafted summaries for the cross-section; the model *learns* the temporal dynamics but the cross-sectional features are hand-designed. For example, Sadhwani et al. (2020) employs aggregate foreclosure rate as a summary feature across the loans. This approach requires domain expertise and is unlikely to capture all latent effects.

To address this two-dimensional challenge, we propose the Set-Sequence model, which decouples the modeling of cross-sectional and temporal dependencies by *learning* latent cross-sectional effects directly from data. The architecture consists of two components (see Figure 1). First, a *Set module* processes the permutation-invariant cross-section at each time step, leveraging unit *exchangeability*[1] to compute an order-invariant summary of the population state. Second, this summary is concatenated to each unit's features and passed to a *Sequence module* (e.g., Transformer or RNN) that models temporal dynamics. The Set-Sequence model fulfills three key desiderata: inference over a variable number of units, handling unaligned units with different start and end times, and integration with standard sequence models. This simple architecture yields significant practical benefits: it removes

---

[1]Exchangeability implies that unit identities (e.g., stock tickers) are irrelevant—behavior is determined solely by observed features. Formally, for a problem of size $M$, the unit-level feature vectors $X^i$ are exchangeable: $P(X^1, \ldots, X^M) = P(X^{\pi(1)}, \ldots, X^{\pi(M)})$ for any permutation $\pi$.

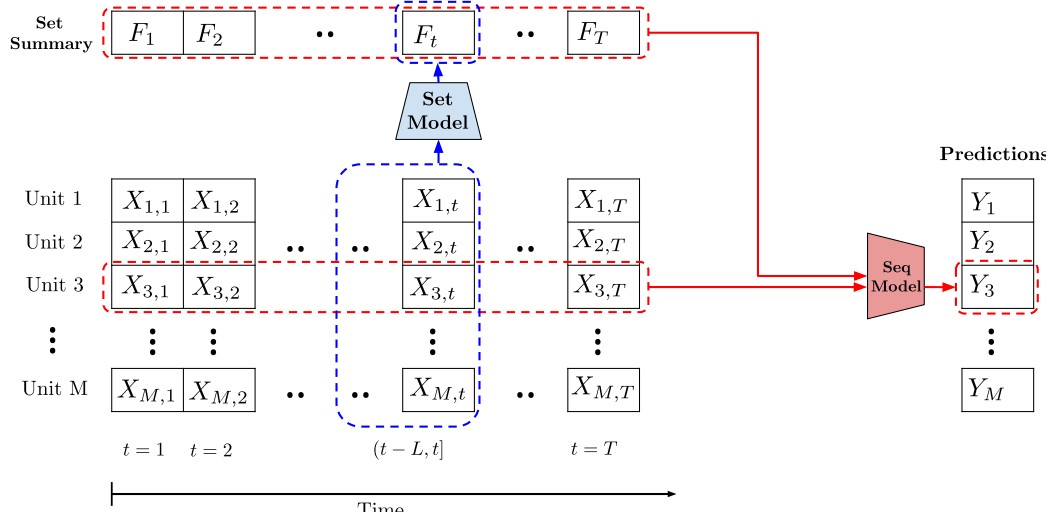

Figure 1: **The Set-Sequence model.** The Set model estimates a cross-sectional summary at each time, in linear complexity over the number of units. This augments the original features of each unit and the Sequence model consumes this augmented series to make predictions for each unit independently. Note that the Set model has a look-back of $L$ time periods, where $L \geq 1$ is a model parameter. The number of units $M$ and time periods $T$ may vary at inference.

the need for handcrafted features and seamlessly incorporates cross-sectional dependencies into unit-level predictions.

Theoretical analysis highlights key properties of the Set-Sequence model with direct practical implications. First, a forward-pass complexity result shows that mean-based cross-sectional aggregation scales *linearly* in the number of units $M$, in contrast to the *quadratic* scaling of attention-based methods—enabling efficient training on large cross-sections. Second, under exchangeability, a polynomial sufficiency result guarantees that pooled monomials up to degree $k$ can approximate any continuous permutation-invariant cross-sectional statistic. Together, these results motivate the use of simple set summaries as a scalable and expressive default, capable of capturing rich latent structure without incurring prohibitive computational cost.

We first benchmark the performance and efficiency of the Set-Sequence model on a synthetic contagion task that mimics default prediction for a large pool of exchangeable loans. Defaults are driven by a shared latent factor $\lambda_t$, capturing contagion effects observed in corporate and consumer credit markets. The Set-Sequence model substantially outperforms state-of-the-art sequence models, achieving near-optimal performance across a wide range of pool sizes, with accuracy improving as coverage increases. It also exhibits strong interpretability: the learned set summaries closely track the true latent factor ($\rho = 0.95$). Notably, the simple linear aggregation used in the Set module outperforms more complex multi-head attention layers, both in accuracy and efficiency.

We next evaluate the model on a real-world stock portfolio construction task using return data for 36,600 U.S. equities. The goal is to select daily portfolio weights that maximize the annualized Sharpe ratio (a widely used performance criterion). We compare Set-Sequence against leading sequence models and the task-specific CNN-Transformer of Guijarro-Ordonez et al. (2022). Over a 15-year out-of-sample period, Set-Sequence achieves 22–42% higher Sharpe ratios than the strongest baselines. These improvements translate into economically meaningful investment gains.

Finally, we apply Set-Sequence to a large-scale U.S. mortgage risk prediction task, using 5 million loan-month observations and 52 dynamic loan-level and macroeconomic features. The task involves classifying the next-month mortgage state (e.g., current, 30/60/90 days delinquent, foreclosure, prepayment). By effectively capturing latent cross-unit dependencies, Set-Sequence achieves significantly higher AUCs than both general-purpose sequence models and the domain-specific deep

learning benchmark of Sadhwani et al. (2020). These gains translate into more accurate risk metrics and improved decision support for large-scale mortgage portfolio management.

This work highlights the value of exploiting exchangeability in high-dimensional time series and demonstrates that the Set-Sequence model offers a practical and effective solution. It: (1) *Outperforms strong baselines* on both synthetic and real-world tasks; (2) *Scales efficiently* to large cross-sections, with linear complexity in the number of units and support for a variable number of units at inference; and (3) *Provides interpretability*, with low-dimensional set summaries that capture and reveal latent cross-sectional structure. Together, these properties make Set-Sequence a compelling architecture for modern time-series prediction problems involving large populations of interacting units.

## 2 RELATED LITERATURE

Traditional methods for modeling multiple time series include Vector Autoregressive (VAR) models Sims (1980), which scale quadratically in the cross-sectional dimension. Linear factor models Chamberlain & Rothschild (1983); Stock & Watson (2002a; 2011) reduce dimensionality but struggle with non-linear dependencies. Deep learning-based factor models Wang et al. (2019) address this limitation, yet require retraining when new units are introduced.

Recent deep learning architectures for time series, though powerful, are typically not designed to exploit cross-sectional exchangeability as they rely on the specific identities of the units and do not scale to arbitrary units at inference. Models like iTransformer Liu et al. (2024) and TimeMixer Wang et al. (2024a) focus on fixed variates or multiscale temporal patterns, respectively. Some approaches model cross-sectional exchangeability more directly, using Gaussian Copulas Salinas et al. (2019) or low-rank matrix factorization Sen et al. (2019), but these impose restrictive assumptions or require refitting when units change. Permutation-invariant architectures such as Deep Sets Zaheer et al. (2017) and Set Transformers Lee et al. (2019) model unordered collections effectively, but are designed for static sets and lack temporal structure. They can handle a set of sequences, but our setting contains additional structure: the sequences live on a shared time axis. Our work introduces a novel dynamic architecture that applies set-based modeling *at each time step*, enabling scalable and expressive handling of high-dimensional, time-indexed cross-sections.

Graph Neural Networks Scarselli et al. (2009) offer a flexible framework for modeling cross-sectional dependencies, especially when an explicit relational graph is available. However, in large-scale applications such as mortgage risk prediction—with hundreds of thousands of loans—inferring a reliable dependency graph is challenging and computationally costly, even if the graph is learned as part of the objective Wu et al. (2020b). In contrast, the Set-Sequence model sidesteps this requirement by learning latent cross-sectional structure directly from observed features.

General-purpose models—such as Transformer variants, State Space Models (S4, H3), and MLP-based architectures Vaswani et al. (2017); Gu et al. (2022); Fu et al. (2022); Chen et al. (2023)—scale quadratically in the number of units, often requiring per-unit modeling with shared weights and handcrafted features to capture joint effects Sadhwani et al. (2020). The Set-Sequence model instead learns cross-sectional dependencies directly, with linear scaling and no feature engineering.

We provide a more in-depth literature review in Appendix B.

## 3 SET-SEQUENCE MODEL

Consider a dataset $\mathcal{D} = \{(Y^i, X^i)\}_{i=1}^{M}$ with $M$ units, where $Y^i = (Y_1^i, Y_2^i, \ldots, Y_T^i) \in \mathbb{R}^{d_y \times T}$ is the response and $X^i = (X_1^i, X_2^i, \ldots, X_T^i) \in \mathbb{R}^{d_x \times T}$ is a covariate. We allow data with different start and end dates but we assume the data are on a regular grid and padded so that all series have the same length. We use the notation $X_{(s,t)}^i = (X_s^i, X_{s+1}^i, \ldots, X_t^i)$ to denote the covariates from time $s$ to $t$ for unit $i$. Given a set of covariates $\{X_{(s,t)}^j\}_{j=1}^{M}$, the goal is to predict $Y_{t+1}^i$.

The Set-Sequence layer combines a set network for cross-sectional data processing with an arbitrary sequence-to-sequence layer for modeling temporal dependencies, as shown in Figure 1.

**Set-Sequence Layer**: Let $X^1, \ldots, X^M$ are the input units to the Set-Sequence layer, and $Y^1, \ldots, Y^M$ are the outputs of the layer. We use temporal chunks as many temporal patterns depend on several time steps to process. These inputs are processed in temporal chunks of size $L$, to allow the summary at the current period to use information from recent previous periods, producing feature

embeddings via an embedding network, $\phi$. These embeddings are then fed through a set network $\rho$ to produce a permutation invariant, low dimensional, set summary $F_t$

$$F_t = \rho\Big(\frac{1}{M}\sum_{i=1}^{M}\phi\big(X_{(t-L,t)}^i\big)\Big) \in \mathbb{R}^r, \tag{1}$$

We then concatenate the set summary to create an augmented feature $\tilde{X}^i$ that is fed individually into a given sequence layer SeqLayer

$$\tilde{X}_t^i = \psi([X_t^i, F_t]^T) \tag{2}$$

$$Y_{t+1}^i = \text{SeqLayer}(\tilde{X}_{(1,t)}^i), \tag{3}$$

for all $i \in \{1, \ldots, M\}$, and $t \in \{1, \ldots, T\}$.

**Set-Sequence Model**: The Set-Sequence model replaces a subset of the layers in a sequence model with the Set-Sequence layer. For example, a Transformer, Long Convolution, SSM, or RNN layer can be used for SeqLayer.

**Alternative Layer Formulations**: We use the linear set aggregation in Equation 1 due to its simplicity and empirically strong performance. As an alternative, we propose MHA-Seq, a multi-head attention layer that scales quadratically with the number of units. This layer is effective when the cross-sectional dimension is small or when cross-unit dependencies are particularly strong. For each unit, we compute a low dimensional set summary by using multi-head attention, by cross attending to all the other units, by the following equation

$$F_t^i = \text{MHA}\left(\big[\phi(X_{(t-L,t)}^1), \ldots, \phi(X_{(t-L,t)}^M)\big]^T\right)_i. \tag{4}$$

Other architectures are explored in Appendix C.

Unless noted otherwise, we use LongConv Fu et al. (2023) for the sequence layer, two-layer FFNs with dropout for $\phi$, $\rho$, and $\psi$, and chunk size L = 3. MHA-Seq differs only by using five attention heads. Full hyperparameters are in Appendix D.

Proposition 1 highlights the improved time complexity of the forward pass of the Set-Sequence model compared with other approaches to model the cross-section. The proofs of this and other results in this section are given in Appendix A.

**Proposition 1** (Forward time for one cross-sectional layer). *Let $M$ be the number of units, $d$ the number of features per unit, and $T$ the sequence length. At each time step $t \in \{1, \ldots, T\}$ we hold $M$ vectors $x_t^{(i)} \in \mathbb{R}^d$. Let $C_{\text{seq}}(T, w)$ denote the cost of one temporal layer on a length-$T$ sequence of width $w$. The forward-time complexity of one cross-sectional layer is:*

| Method | Time Complexity |
|---|---|
| Set–Seq | $\Theta(TMd) + \Theta\big(M\,C_{\text{seq}}(T, d)\big)$ |
| MHA–Seq | $\Theta\big(T[M^2d + Md^2]\big) + \Theta\big(M\,C_{\text{seq}}(T, d)\big)$ |
| Naïve MHA–Seq | $\Theta\big(T(Md)^2\big) + \Theta\big(M\,C_{\text{seq}}(T, d)\big)$ |
| Full-Stacked Temporal | $\Theta\big(C_{\text{seq}}(T, Md)\big)$ |

*MHA–Seq applies cross-sectional self-attention across the $M$ units, whereas the Naïve MHA–Seq instead treats each feature as a token ($N = Md$). Full-Stacked Temporal applies a single joint temporal model directly to the concatenated $Md$-dimensional features.*

**Expressivity under exchangeability**: When the cross-section at each time is exchangeable—i.e., predictions depend on the *multiset* of unit features rather than their identities—using a permutation-invariant set summary does not sacrifice expressivity for continuous targets on compact domains. The result below formalizes this: pooled monomial features up to a sufficiently large degree $k$ suffice to approximate any continuous permutation-invariant cross-sectional function.

**Proposition 2** (Expressivity of Set Module). *Let $K \subset \mathbb{R}^d$ be compact and fix $M \in \mathbb{N}$. Let $G : K^M \to \mathbb{R}^q$ be continuous and permutation-invariant, i.e., $G(x^{(\pi(1))}, \ldots, x^{(\pi(M))}) = G(x^{(1)}, \ldots, x^{(M)})$ for every permutation $\pi$. Then for every $\varepsilon > 0$ there exist an integer $k = k(\varepsilon, G, K, d, M) \geq 0$, a*

*feature map $\phi_k : K \to \mathbb{R}^{\binom{d+k}{k}}$ containing all monomials of total degree $\leq k$, and a continuous map $\rho : \mathbb{R}^{\binom{d+k}{k}} \to \mathbb{R}^q$ such that*

$$\sup_{(x^{(1)},\ldots,x^{(M)}) \in K^M} \left\| \rho\Big( \tfrac{1}{M} \sum_{i=1}^{M} \phi_k(x^{(i)}) \Big) - G(x^{(1)},\ldots,x^{(M)}) \right\|_\infty < \varepsilon.$$

*Equivalently, the set module $F = \rho\big(\tfrac{1}{M} \sum_{i=1}^{M} \phi_k(x^{(i)})\big)$ uniformly approximates $G$ on $K^M$.*

## 4 TASK SELECTION AND BASELINES

**Task Selection**: Common multivariate-forecasting benchmarks—ETTh1, ETTm1, Exchange, and Traffic (see details in Zhou et al. (2021))—are ill-suited for our study: their unit counts are small enough that a full joint model remains tractable or they lack a clear exchangeable-unit structure. We therefore evaluate on three datasets—a synthetic contagion setting, an equity-portfolio regression task, and a mortgage-risk classification task—each defined over exchangeable units. They contain 1000, 500, and 2500 units with 4, 79, and 52 features per unit, giving cross-sectional dimensions of 4000, 39500, and 130000 that meaningfully stress models built for high-dimensional cross-sections.

**General and Domain-Specific Baselines**: We benchmark against five widely adopted sequence architectures— Transformer Vaswani et al. (2017), S4 Gu et al. (2022), H3 Fu et al. (2022), Hyena Poli et al. (2023), and LongConv Fu et al. (2023). Together they span attention-, state-space-, and convolution-based methodologies, each of which our Set-Sequence layer can augment. We treat them as task-agnostic baselines on all three datasets. Because these models share a common backbone structure, we hold input/output dimensions, hidden size, learning-rate schedule, and other hyperparameters fixed, so the only variation lies in the sequence layer (and whether a Set module is present). This yields a fair comparison across models. Each task section also reports results for the strongest task-specific domain baseline. Hyperparameters for all models are in Appendix D.

## 5 SYNTHETIC TASK

We examine the Set-Sequence model's performance on a synthetic task mimicking loan default prediction, where contagion is a key latent factor Anenberg & Kung (2014); Azizpour et al. (2018); Towe & Lawley (2013). The goal is to predict the next-step transition probabilities for each unit.

We simulate 1000 exchangeable units, each with binary feature $x \in \{0, 1\}$ over 100 time steps. Units transition between three states (state 3 is absorbing/default) with a transition matrix proportional to:

$$\begin{pmatrix} 1+x & 1 & (\lambda_{x,t} + \mu)(1 + 0.1x) \\ 1 & 1+x & (\lambda_{x,t} + \mu)(1 + 0.1x) \\ 0 & 0 & 1 \end{pmatrix} \tag{5}$$

The default rate evolves as $\lambda_{x,t+1} = \beta \lambda_{x,t} + \alpha N_{x,t}$, where $N_{x,t}$ is the fraction of type-$x$ units defaulting at time $t$. Parameters $\mu = 0.001$, $\alpha = 4$, and $\beta = 0.5$ yield an average default rate of $\sim 1\%$. The model captures contagion via $\lambda_t = (\lambda_{0,t}, \lambda_{1,t})$, where defaults increase future default risk. The core challenge is to model the joint dependencies created by the latent factor $\lambda_t$, which is itself a function of each unit's state.

### 5.1 COMPARISON WITH SEQUENCE MODELS

We compare the Set-Sequence model to the sequence baselines from Section 4 using AUC for the rare default transition and KL divergence between true and predicted transition probabilities. Table 1 reports KL and AUC for each backbone in three modes: *Joint* (a single sequence over the full cross-section), *Single* (one unit at a time), and *Set–Seq* (our set-summary + sequence backbone). Joint models underperform, highlighting the difficulty of modeling large cross-sections with a sparse joint signal. Moving from Joint to Single recovers substantial accuracy, and adding the Set component yields the best results across all backbones: KL improves by $4.4\times$ (H3) to $10.2\times$ (LongConv), with AUC gains of about $+0.04$ in each case, indicating that the benefit of the Set summary is robust to the choice of sequence backbone. In later experiments we use the LongConv sequence model backbone.

**Model Ablations**: Table 2 reports the results of ablation study, varying the cross-sectional component (None, Set, or MHA) and sequence length. The largest performance gain comes from adding the

Table 1: Backbone-agnostic evaluation of adding Set–Seq. For each backbone we report KL (↓) and AUC (↑) in three modes: *Joint* (concatenate all unit features and model a single sequence over the full cross-section), *Single* (model each unit independently), and *Set–Seq* (set summary + per unit backbone; values in **bold**). The rightmost column gives the multiplicative KL reduction of Set–Seq relative to Single.

| Backbone | KL Joint ↓ | AUC Joint ↑ | KL Single | AUC Single | KL Set–Seq | AUC Set–Seq | KL× (vs Single) |
|---|---|---|---|---|---|---|---|
| LongConv | 0.037 | 0.681 | 0.0018 | 0.757 | **0.00018** | **0.802** | 10.2 |
| S4 | 0.038 | 0.676 | 0.0016 | 0.758 | **0.00019** | **0.803** | 8.1 |
| H3 | 0.040 | 0.675 | 0.0017 | 0.751 | **0.00039** | **0.795** | 4.4 |
| Transformer | 0.036 | 0.506 | 0.0016 | 0.758 | **0.00021** | **0.801** | 7.6 |
| Hyena | 0.036 | 0.702 | 0.0017 | 0.760 | **0.00019** | **0.802** | 9.1 |

Table 2: Ablation study of the Set-Sequence model on the synthetic task, varying the cross-sectional modeling (None, Set, or Multi-Head Attention) and sequence length (1 or 50). Sequence Length refers to the model's effective lookback or kernel length. Relative Epoch Time and Relative Max Train Memory are reported relative to the Set-Sequence model with Set and Sequence Length 50. All models were trained on an Nvidia RTX A6000 GPU.

| Seq. Model | Set Model | Seq. Len. | KL(true\|predicted) | AUC | Rel. Epoch Time | Rel. Mem |
|---|---|---|---|---|---|---|
| LongConv | None | 1 | 0.0021 | 0.693 | 0.51 | 0.86 |
| LongConv | None | 50 | 0.0018 | 0.757 | 0.64 | 0.93 |
| LongConv | Set | 1 | 0.00054 | 0.792 | 0.60 | 0.88 |
| LongConv | Set | 50 | 0.00018 | 0.802 | 1.00 | 1.00 |
| LongConv | MHA | 50 | 0.000044 | 0.815 | 3.35 | 3.52 |

Set component. While replacing it with Multi-Head Attention (MHA) further improves accuracy, it incurs a $3.3\times$ training time and $3.5\times$ memory increase due to its quadratic scaling.

### 5.2 GENERALIZATION ACROSS NUMBER OF INPUT UNITS

The task is to accurately predict state transitions when only a fraction $n$ of the total $M$ units are observed at inference time. This simulates having a large, evolving universe of loans but only observing a limited subset. We develop a near-optimal Kalman Filter baseline that assumes the true transition dynamics are known, with uncertainty only arising from observing a fraction of the total defaults. This oracle-like baseline represents a performance upper bound for the task. The full derivation is in Appendix E.3. During training we draw the number of units per batch as $D \sim (1 - \gamma) \cdot \delta_M + \gamma \cdot \text{LogUnif}(1, M)$ with $\gamma = 0.08$; thus most batches use all $M = 1000$ units, while a few use smaller pools to encourage generalization. Further details are in Appendix E.

We evaluate the model's ability to predict transitions to the absorbing (rare) state using AUC, correlation, and $R^2$. Formal definitions are in Appendix E. Figure 2 shows that both the Set-Sequence and MHA-Seq models generalize well across all inference sizes, with performance approaching the oracle-like Kalman Filter baseline even when trained on a finite dataset. A single model effectively handles input sizes from 1 to 1000 units. While the MHA-Seq variant performs slightly better, it comes at a significant computational cost due to its quadratic scaling. The Set-Sequence model is interpretable. Table 3 shows a high correlation (up to 0.951) between a learned set summary and the true latent factor $\lambda_{0,t}$, with the correlation increasing as more units are observed. Appendix E.2 contains visualizations.

## 6 APPLICATION: EQUITY PORTFOLIO CONSTRUCTION

We evaluate the Set-Sequence model on a portfolio construction task. Given a universe of $N$ stocks, at the current time, the model outputs $N$ portfolio weights, indicating the portfolio to be held. The portfolio returns are realized from the inner product between the next day returns and the chosen

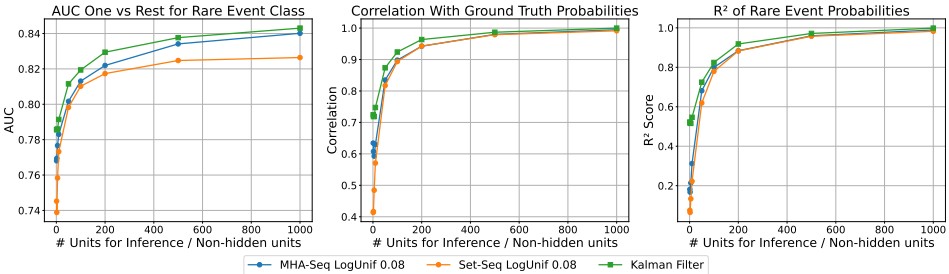

Figure 2: Comparison between the Set-Sequence model, MHA-Sequence model, and the Kalman Filter, by considering the AUC, $R^2$, and Correlation, each for the absorbing (rare) state. The Set-Sequence model reaches near the oracle-like Kalman Filter model performance across the full range of observed units for inference, for all the considered metrics.

Table 3: Average correlation for 100 samples, each with 100 time steps between the set summary at layer $i$ and the true latent variable, $\lambda_{0,t}$. Here and in subsequent tables, best (second best) values in each column are in bold (underlined).

| # obs | 20 | 50 | 100 | 200 | 500 | 1000 |
|---|---|---|---|---|---|---|
| Layer 1 | 0.234 | 0.237 | 0.236 | 0.233 | 0.233 | 0.232 |
| Layer 2 | 0.160 | 0.165 | 0.165 | 0.167 | 0.165 | 0.166 |
| Layer 3 | 0.349 | 0.478 | 0.576 | **0.699** | **0.860** | **0.951** |
| Layer 4 | 0.263 | 0.330 | 0.414 | 0.547 | 0.642 | 0.690 |
| Layer 5 | **0.444** | **0.520** | **0.610** | 0.681 | 0.759 | 0.804 |

portfolio weights. The goal is to optimize the Sharpe ratio, a measure of the risk-adjusted returns. This is a common objective in the asset pricing literature, as in Guijarro-Ordonez et al. (2022).

**Dataset**: We construct a dataset of 36,600 stocks (source: CRSP). For each period, we train on 8 years and test the model on the following year. For each model fitting, we filter out stocks where any returns in the train or test period are not available. We include only the 500 assets with the largest market cap at the last quarter of the training period, mimicking the S&P 500 index. This ensures that we consider assets that are liquid with small bid-ask spreads, see Guijarro-Ordonez et al. (2022). We use 79 features from Chen et al. (2024), applying cross-sectional rank normalization. They are based on recent cumulated returns, volatility, volume, and quarterly firm characteristics (such as book-to-market ratio). See Tables 10 and 11 in Appendix F.

**Objective**: Let $X_t^i$ be the covariates at time $t$, and let $Y_t^i$ be the excess daily returns for asset $i$ (returns with the risk free rate subtracted), where $X_t$ is the covariate vector, and $Y_t$ is the return vector. We then use $X_1, \ldots, X_{t-1}$ to predict $\hat{w}_t$, the portfolio weights at the next step. Given a series of daily excess returns, $r_1, \ldots, r_T$, where $r_t = Y_t^T \hat{w}_t$, the daily Sharpe ratio is $SR = \bar{r} / \sqrt{\frac{1}{T} \sum_{t=1}^{T} (r_t - \bar{r})^2}$ where $\bar{r} = \frac{1}{T} \sum_{t=1}^{T} r_t$. We $L_1$ normalize the portfolio weights at each timestep to maintain unit leverage ($||w_t||_1 = 1$). Following Guijarro-Ordonez et al. (2022) we optimize $-SR$.

**Baselines, Metrics and Results**: We compare with the sequence baselines in Section 4. Each baseline is trained on a single unit at a time with weights shared across units—this generalizes far better than feeding the entire cross-section, which performs poorly (see Section 5.1). We further compare with task-specific models: the CNN-Transformer and the baselines of Guijarro-Ordonez et al. (2022), which use aligned dataset, time span, and feature set (following Chen et al. (2024)).

The main metric to evaluate the performance is the annualized Sharpe ratio. We also report the percentage yearly return, the yearly return volatility, the average daily turnover, defined as the average $L_1$ distance between the weights in consecutive days. Beta is the covariance of portfolio and market returns divided by the market variance, measuring market exposure. A low beta is desirable to ensure returns are independent of market direction.

Table 4: Summary statistics for the equities task out of sample (Jan. 2002–Dec. 2021). Each model is trained five times on different random seeds; all values are the mean over those runs, and the Sharpe ratio is reported as mean ± std. The Sharpe Ratio, Mean Return, and Std-Dev Return are annualized. Beta is relative to the market, while Daily Turnover and Short Fraction are daily averages.

| Model | Sharpe Ratio | Return % | Std Dev % Return | Beta | Daily Turnover | Short Fraction |
|---|---|---|---|---|---|---|
| LongConv | $3.64 \pm 0.14$ | 12.8 | 3.51 | 0.033 | 0.97 | 0.48 |
| S4 | $\underline{3.94} \pm 0.29$ | 13.5 | 3.43 | 0.028 | 0.90 | 0.48 |
| H3 | $3.46 \pm 0.28$ | 10.6 | 3.08 | 0.026 | 1.12 | 0.48 |
| Transformer | $3.65 \pm 0.52$ | 12.9 | 3.62 | 0.035 | 0.83 | 0.47 |
| Hyena | $2.91 \pm 0.33$ | 8.7 | 3.02 | 0.032 | 1.28 | 0.47 |
| Set-Sequence (Ours) | $\mathbf{4.82} \pm 0.12$ | 13.0 | 2.69 | 0.028 | 0.91 | 0.48 |

Table 4 shows the aggregate performance. The Set-Sequence model outperforms every sequence model baseline. We note a high annualized mean Sharpe ratio over the full period of 4.82, 22% higher than the second best sequence model (S4), and 32% higher than the LongConv model, which the Set-Sequence model uses for the sequence component in the experiment. Compared with the other sequence models, the Set-Sequence model shows robust results, with a Sharpe ratio standard deviation over 5 random initialized training runs of 0.12, compared with 0.52 for the Transformer and 0.29 for S4. Figure 8 in the Appendix shows the cumulative returns for the models.

We also compare the Set-Sequence model with domain specific models on the time period Jan. 2002 - Dec. 2016 to be aligned with time period of prior work. Table 5 reports that the general Set-Sequence model outperforms the CNN-Transformer designed specifically for the equities task by 42 % in terms of the Sharpe ratio. In Section F in the Appendix we include an analysis of the Sharpe ratio when we account for *transaction costs*. We use a net Sharpe objective function and show that the Set-Sequence model still outperforms all baselines in the presence of transaction costs.

Table 5: Out-of-sample annualized performance metrics from 2002 to 2016 for the stock portfolio construction task. All CNN-Transformer models are based on the IPCA method described in Guijarro-Ordonez et al. (2022). For the Set-Sequence model the results are the mean over 5 training initialization seeds. For the Sharpe we also report its standard deviation over the seeds.

| Model | Sharpe | $\mu$ [%] | $\sigma$ [%] |
|---|---|---|---|
| CNN+Trans K=5 | 4.16 | 8.7 | 2.1 |
| CNN+Trans K=8 | 3.95 | 8.2 | 2.1 |
| CNN+Trans K=10 | 3.97 | 8.0 | 2.0 |
| CNN+Trans K=15 | $\underline{4.17}$ | 8.4 | 2.0 |
| Fourier+FNN K=10 | 1.93 | 7.6 | 3.9 |
| Fourier+FNN K=15 | 2.06 | 7.9 | 3.8 |
| OU+Thresh K=10 | 0.86 | 3.1 | 3.6 |
| OU+Thresh K=15 | 0.93 | 3.2 | 3.5 |
| Set-Seq (Ours) | $\mathbf{5.91} \pm 0.21$ | 14.7 | 2.5 |

Table 6: Comparing out-of-sample performance of different models on the mortgage risk prediction task on the test set. The average AUC is the mean of the AUCs to go between different states. The average transition probabilities are the average probability for the correct class over the full test set.

| Model | Cross Entropy | Avg. AUC |
|---|---|---|
| 5-Layer NN | $\underline{0.205}$ | 0.642 |
| Log. regression | 0.225 | 0.622 |
| LongConv | 0.216 | 0.669 |
| S4 | 0.226 | $\underline{0.681}$ |
| H3 | 0.222 | 0.583 |
| Transformer | 0.227 | 0.666 |
| Hyena | 0.213 | 0.674 |
| Set-Seq (Ours) | $\mathbf{0.200}$ | $\mathbf{0.683}$ |

## 7  APPLICATION: MORTGAGE RISK PREDICTION

Next we evaluate the Set-Sequence model on an important mortgage risk prediction task, with the goal to predict the mortgage state (current, 30 days delinquent, 60 days delinquent, 90+ days delinquent, Foreclosure, Paid Off, and Real-Estate Owned) in the next month given a history of 52 covariates (FICO credit score, loan balance, current interest rate, etc., see Table 12 in the Appendix).

**Dataset**: We source the data from 4 ZIP codes in the greater LA area from CoreLogic; these are among the ZIP codes in the US with most active mortgages. We follow the data filtering procedure of

Sadhwani et al. (2020), resulting in 5 million loan-month transitions from 117,523 loans. We use the following train, validation, test split: 1/1994 - 6/2009, 7/2009 - 12/2009, 1/2010 - 12/2022.

**Baselines, Metrics and Results**: We compare the Set-Sequence model with the domain specific models presented in Sadhwani et al. (2020), representing the current state of the art, along with the sequence models from Section 4. The models from Sadhwani et al. (2020) include a 5-layer neural network baseline with early stopping and dropout regularization, and a logistic regression baseline. Both baselines use the features at the current time to predict the state in the following month. For each training batch, for a given time we sample 2500 loans out of all active ones. We use a sequence length of 50, and use a multi-class cross entropy loss during training.

Table 6 shows that the Set-Sequence model achieves the best performance on cross-entropy loss and average AUC, outperforming all baselines. In particular, it improves average AUC by 4 points over the best domain-specific baseline (5-layer neural network). In Figure 3 we see that the Set-Sequence model robustly outperforms the prior state of the art model on the task, with better AUC for 22 out of 25 transitions, including the most common and economically important transitions, such as current to paid off. In Figure 12 in the Appendix, the foreclosure rate, a known source of cross unit dependency, is shown to be highly correlated with the set summary in the first Set-Sequence layer, indicating the interpretability of the learned set summaries.

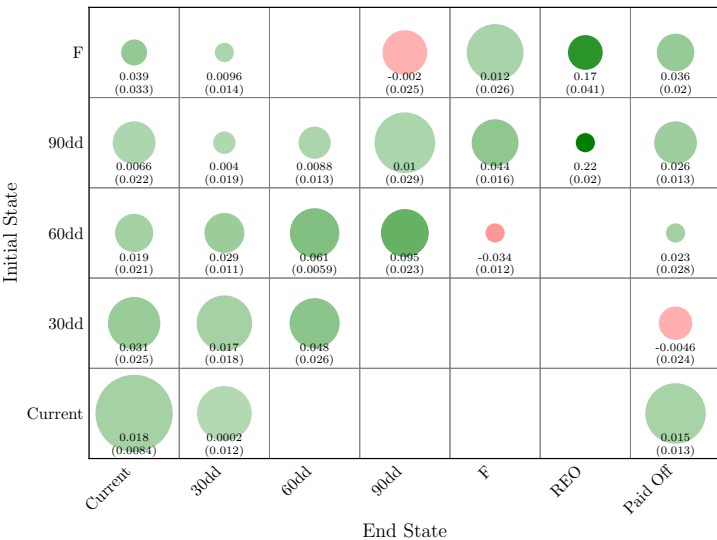

Figure 3: AUC gain (Set-Sequence minus baseline Sadhwani et al. (2020)) for each transition. Deeper green indicates a larger Set-Sequence advantage; red indicates the baseline performed better. Larger circles indicate more common transitions, with size proportional to the log of their frequency. Only transitions occurring at least 10 times are included. In parentheses is the standard deviation over 10 random subsets of the evaluation set.

Appendix G supplements the discussion above with additional details and results. These include additional experiments for year-by-year performance and additional interpretability results.

## 8 CONCLUSION

This paper introduced the Set-Sequence model, an efficient architecture for capturing latent cross-sectional dependencies without manual feature engineering. By learning a shared cross-sectional summary at each period with a Set model and subsequently augmenting individual unit time series for a Sequence model, our approach demonstrates strong empirical performance on synthetic, investment, and risk prediction tasks, significantly outperforming benchmarks. While many existing multivariate models are powerful, they are often not natively designed to exploit unit exchangeability in multivariate time series. The Set-Sequence model offers a simple and scalable solution.

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

# A THEORY

## A.1 PROOF OF PROPOSITION 1 AND A COROLLARY

*Proof.* We count scalar operations to within constant factors and then sum over $t = 1, \dots, T$.

**Preliminaries.** At any time $t$, forming per-unit embeddings/fusions by dense linear maps on $x_t^{(i)} \in \mathbb{R}^d$ costs $\Theta(d)$ per unit, hence $\Theta(Md)$ across all units. Mean/sum pooling of $M$ vectors in $\mathbb{R}^d$ is a single pass over $Md$ scalars, i.e. $\Theta(Md)$.

For cross-sectional attention over $M$ tokens of width $d$, the standard forward pass per head entails: (i) Q/K/V linear maps: $\Theta(Md^2)$, (ii) score matrix $QK^\top$: $\Theta(M^2 d)$, (iii) value aggregation $(\text{softmax}(QK^\top))V$: $\Theta(M^2 d)$, (iv) output projection: $\Theta(Md^2)$, and an FFN of width $\Theta(d)$ per token: $\Theta(Md^2)$. Grouping linear terms yields $\Theta(Md^2)$ and the pairwise terms yield $\Theta(M^2 d)$. Multi-head with a constant number of heads only changes constants.

When each scalar becomes a token (Naïve feature-as-token), there are $N = Md$ tokens. With width $\Theta(1)$ per token, Q/K/V/out projections are $\Theta(N)$ and attention multiplies are $\Theta(N^2) = \Theta((Md)^2)$, which dominates.

**(i) Set–Seq.** At each $t$, compute per-unit transforms and fuse with the pooled summary. This is one linear scan over all $M$ units with $d$ features: $\Theta(Md)$ to embed, $\Theta(Md)$ to pool, and $\Theta(Md)$ to fuse; thus $\Theta(Md)$ up to constants. Summed over $T$ time steps gives $\Theta(TMd)$ for the cross-sectional part. Then we run one temporal model per unit on width $d$, for a total of $\Theta\big(M\, C_{\text{seq}}(T, d)\big)$. Hence (i).

**(ii) Attention–Seq (M tokens, width d).** Per $t$, cross-sectional attention over $M$ tokens of width $d$ costs $\Theta(M^2 d)$ for scores/aggregation and $\Theta(Md^2)$ for projections/FFN, as detailed above. Summed over $T$ time steps gives $\Theta(T[M^2 d + Md^2])$. Temporal processing is still per unit at width $d$, adding $\Theta\big(M\, C_{\text{seq}}(T, d)\big)$. Hence (ii).

**(iii) Naïve feature-as-token (N=Md).** Per $t$, there are $N = Md$ tokens. With token width $\Theta(1)$, the attention-matrix multiplies (scores and aggregation) cost $\Theta(N^2) = \Theta((Md)^2)$ and dominate the $\Theta(N)$ projection costs. Summed over $T$ gives $\Theta(T(Md)^2)$. Temporal processing remains per unit at width $d$, adding $\Theta\big(M\, C_{\text{seq}}(T, d)\big)$. Hence (iii).

**(iv) Gated-Selection–Seq.** By assumption, the cross-sectional gating/selection matrix is computed *once* (not per time) from $M$ unit summaries of width $d$ via a dense similarity, which is $\Theta(M^2 d)$: there are $\Theta(M^2)$ pairs and computing each affinity costs $\Theta(d)$. Thereafter, at each time step the layer behaves as Set–Seq (no pairwise recomputation), costing $\Theta(Md)$ per $t$, hence $\Theta(TMd)$ across the sequence, plus the per-unit temporal total $\Theta\big(M\, C_{\text{seq}}(T, d)\big)$. Hence (iv).

**(v) Full-Stacked Temporal.** At each $t$, stack all $M$ unit features into a single vector of width $Md$ (reshaping costs $O(Md)$ and is dominated by the temporal block). Apply a single temporal model of width $Md$ over length $T$, whose cost is by definition $C_{\text{seq}}(T, Md)$. No per-unit temporal pass occurs. Hence (v).

**Conclusion.** Combining the per-time costs (and the one-off gate in (iv)) with the temporal costs yields the stated asymptotics. Backpropagation traverses the same computations with constant-factor overhead, leaving the $\Theta(\cdot)$ orders unchanged. This completes the proof. $\square$

**Corollary 1** (Plugging in standard temporal self-attention)**.** *If the temporal model is standard self-attention with cost* $C_{\text{attn}}(T, w) = \Theta(T^2 w) + \Theta(Tw^2)$*, then:*

$$
\begin{aligned}
\text{\it (i) Set–Seq:} \quad & \Theta(TMd) \;+\; \Theta\big(M[T^2 d + Td^2]\big), \\
\text{\it (ii) X-section Attn (M tokens):} \quad & \Theta\big(T[M^2 d + Md^2]\big) \;+\; \Theta\big(M[T^2 d + Td^2]\big), \\
\text{\it (iii) Naïve (Md tokens):} \quad & \Theta\big(T(Md)^2\big) \;+\; \Theta\big(M[T^2 d + Td^2]\big), \\
\text{\it (iv) Gated-Selection–Seq:} \quad & \Theta(M^2 d) \;+\; \Theta(TMd) \;+\; \Theta\big(M[T^2 d + Td^2]\big), \\
\text{\it (v) Full-Stacked Temporal:} \quad & \Theta\big(T^2 Md + T(Md)^2\big).
\end{aligned}
$$

## A.2 PROOF OF PROPOSITION 2

*Proof.* **Step 1 (Empirical moments as pooled monomials).** For a multi-index $\alpha = (\alpha_1, \ldots, \alpha_d) \in \mathbb{N}_0^d$ with $|\alpha| = \sum_{j=1}^d \alpha_j$, define $x^\alpha = \prod_{j=1}^d x_j^{\alpha_j}$ and, for any $(x^{(1)}, \ldots, x^{(M)}) \in K^M$, the empirical moment

$$m_\alpha = \frac{1}{M} \sum_{i=1}^M (x^{(i)})^\alpha = \frac{1}{M} \sum_{i=1}^M \prod_{j=1}^d (x_j^{(i)})^{\alpha_j}.$$

Let $\phi_k(x)$ stack all monomials with $|\alpha| \leq k$. Then $\frac{1}{M} \sum_{i=1}^M \phi_k(x^{(i)})$ stacks all $\{m_\alpha : |\alpha| \leq k\}$. The number of such monomials is $\sum_{r=0}^k \binom{d+r-1}{r} = \binom{d+k}{k}$ (stars-and-bars).

**Step 2 (Factorization through the empirical measure; well-defined and continuous).** Define the empirical measure $\mu = \frac{1}{M} \sum_{i=1}^M \delta_{x^{(i)}}$. Any permutation of $(x^{(1)}, \ldots, x^{(M)})$ leaves $\mu$ unchanged and, by invariance, leaves $G$ unchanged. Define

$$\widetilde{G}(\mu) := G(x^{(1)}, \ldots, x^{(M)}) \quad \text{for any ordering that realizes } \mu,$$

which is well-defined because equal empirical measures correspond to permutations of the same multiset and $G$ takes equal values on such permutations. The map $(x^{(1)}, \ldots, x^{(M)}) \mapsto \mu$ is continuous in the weak topology on $K$ (if $x^{(i)} \to y^{(i)}$, then for every continuous $f$ on $K$, $\int f \, d\mu \to \int f \, d\nu$ where $\nu = \frac{1}{M} \sum_i \delta_{y^{(i)}}$). Since $G = \widetilde{G} \circ \mu$ and $G$ is continuous on $K^M$, $\widetilde{G}$ is continuous on the compact image of $K^M$ under this map.

**Step 3 (Polynomial moments separate empirical measures).** On compact $K$, polynomials are dense in $C(K)$ (Stone–Weierstrass). Thus for any two distinct empirical measures $\mu \neq \nu$ there exists a polynomial $p$ with $\int p \, d\mu \neq \int p \, d\nu$. Hence the collection $\{\int p(x) \, d\mu(x) : p \text{ polynomial}\}$ separates empirical measures.

**Step 4 (Approximate $\widetilde{G}$ by a polynomial in finitely many moments).** The domain of $\widetilde{G}$ is compact, and the algebra generated by polynomial moments contains constants and separates points. By Stone–Weierstrass, for any $\varepsilon > 0$ there exists a polynomial $\Psi$ in finitely many moments $\{\int p_\ell \, d\mu\}_{\ell=1}^L$ such that

$$\sup_{(x^{(1)}, \ldots, x^{(M)}) \in K^M} \left\| \Psi(\mu) - \widetilde{G}(\mu) \right\|_\infty < \varepsilon/2.$$

Each $\int p_\ell \, d\mu$ equals $\frac{1}{M} \sum_{i=1}^M p_\ell(x^{(i)})$, and for some degree $k$ each $p_\ell$ is a linear combination of monomials of total degree $\leq k$.

**Step 5 (Realize $\Psi$ via pooled monomials).** With $\phi_k$ as in Step 1, there exists a (multivariate polynomial) map $\rho : \mathbb{R}^{\binom{d+k}{k}} \to \mathbb{R}^q$ such that

$$\Psi(\mu) = \rho\left( \frac{1}{M} \sum_{i=1}^M \phi_k(x^{(i)}) \right) \quad \text{for all } (x^{(1)}, \ldots, x^{(M)}) \in K^M.$$

**Step 6 (Uniform error bound).** Combining Steps 4–5,

$$\sup_{(x^{(1)}, \ldots, x^{(M)}) \in K^M} \left\| \rho\left( \frac{1}{M} \sum_{i=1}^M \phi_k(x^{(i)}) \right) - G(x^{(1)}, \ldots, x^{(M)}) \right\|_\infty \leq \varepsilon/2 < \varepsilon,$$

after tightening constants if desired. This completes the proof. $\qquad \square$

**Remark (Quadratic case).** Taking $k = 2$ and $\phi_2(x) = [\, x, \; \mathrm{vec}_{\mathrm{sym}}(xx^\top) \,]$ recovers first and second empirical moments exactly, so any permutation-invariant quadratic statistic of the cross-section is obtained by a linear $\rho$.

# B ADDITIONAL LITERATURE REVIEW

Many problems exhibit cross-sectional exchangeability: the identity of each unit (e.g., stock, loan) is irrelevant—behavior depends only on observed features. Traditional multivariate time-series models can still be applied in two ways: (i) individual modeling, where one unit is processed at a time with shared parameters, or (ii) a full joint model, which is oftentimes infeasible.

**Individual Unit Modeling**   This approach naturally handles a variable number of units and allows immediate inference for unseen units because the same parameters are shared, but it ignores cross-unit dependencies. Representative methods include classical Vector Autoregressive (VAR) models Sims (1980), which scale quadratically in the cross-section, and a broad class of modern deep-learning time-series models: Transformer variants Zhou et al. (2021); Liu et al. (2022); Zhou et al. (2022); Wu et al. (2021); Piao et al. (2024), state-space models Gu et al. (2022); Zhang et al. (2023); Wang et al. (2025b), convolutional models Fu et al. (2023), multilayer perceptrons Chen et al. (2023); Yi et al. (2023), and graph neural networks Wu et al. (2020a); Shang et al. (2021). Recent time series forecasting models Wang et al. (2025a; 2024b) are complementary in that they can be used as the sequence model backbone for the Set-Sequence model.

**Exchangeability with Fixed Input Size**   These models treat units as exchangeable but must be retrained when new units appear.

1. **Dynamic factor models.** Linear versions Chamberlain & Rothschild (1983); Stock & Watson (2002a; 2011; 2002b); Bok et al. (2018) reduce dimensionality but require re-estimating loadings (and sometimes factors) whenever units change. Deep factor models Wang et al. (2019) replace linear dynamics with RNNs, yet still demand refitting loadings and use local linear/Gaussian components that may miss nonlinear effects.

2. **Joint deep models.** Applying the methods from Paragraph B to the full cross-section produces an input size of ( #units × features/unit), often hundreds of thousands, rendering architectures with quadratic cross-sectional cost impractical.

**Exchangeability with Dynamic Number of Units**   Salinas *et al.* Salinas et al. (2019) use one RNN per unit with shared weights and couple units via a low-rank Gaussian copula; this has only been shown for single-feature units and linear covariance structure. Li *et al.* Sen et al. (2019) combine global matrix factorization with temporal convolutions, but loadings must be refit when units change. Cross-sectional attention models such as iTransformer Liu et al. (2024) and CrossFormer Zhang & Yan (2023) achieve permutation invariance, yet their cost remains quadratic in ( #units × features/unit). In contrast, the Set-Sequence representation uses only #units tokens, enabling efficient joint modelling while building on single-unit sequence architectures.

**Financial Applications**   Joint modelling is vital in finance—for instance, contagion in loan defaults Anenberg & Kung (2014); Azizpour et al. (2018); Towe & Lawley (2013). Previous work often operated on one unit at a time Sadhwani et al. (2020); Khandani et al. (2010). In portfolio optimisation, models typically impose a fixed ordering and limit cross-sectional width Kisiel & Gorse (2023), overlooking exchangeability.

## C   SET-SEQUENCE MODEL WITH GATED SELECTION

This section describes an alternative modeling approach we found to be promising. The *Gated Selection* cross-sectional modeling approach is computationally in between the cost for the full attention and the linear set aggregation in the cross section.

**Gated Selection Layer**   We use temporal chunks as many temporal patterns depend on several time steps to process. These inputs are processed in temporal chunks of size $L$, to allow the summary at the current period to use information from recent previous periods, producing feature embeddings via an embedding network, $\phi$. Then, we define a gating matrix G by the following equations

$$A_{ij} = \frac{X_i^T W^T W X_j}{\|WX_i\|_2 \|WX_j\|_2}$$

$$G = \text{Softmax}(A),$$

where the Softmax is taken along each row. Finally, we produce a per-unit, low-dimensional set summary $F_t^i$ by

$$F_t^i \;=\; \rho\Big( \sum_{j=1}^{M} G_{i,j}\, \phi\big(X_{(t-L,\,t)}^j\big)\Big) \;\in\; \mathbb{R}^r,$$

where $\rho(\cdot)$ is an output projection. The gating, with $G_{i,j} \in [0, 1]$ allows the summary statistic of unit i to mainly depend on the other units with 'similar' features to unit i. We recover the standard Set-Sequence model when we set $G_{i,j} = \frac{1}{M}$, for all $i, j$ where $M$ is the number of units. We provide experiments using the Gated Selection mechanism in Section G.3.3.

**Similarity with Attention**   The Selection mechanism shares important similarities with attention, but is different in the following ways:

- It takes the key and query matrix to be the same (which makes sense given the unit exchangability)

- It replaces the $\sqrt{d}$ normalization with a norms of $WX_i$ and $WX_j$, also motivated by unit exchangeability, where all should have similar norms.

- Most importantly, we only compute one G for all times for the sample, this has a large impact on the memory usage, removing the linear scaling with sequence length needed for gradient computations in the selection mechanism.

## D   EXPERIMENT DETAILS

We here detail the hyperparameters used for training the Set-Sequence model. The hyperparameters for the synthetic task, mortgage risk task, and equity portfolio prediction task are shown in Table 7. In Table 8 we show the layer hyperparameters for the sequence model baselines used. The number of layers, learning rate schedule, learning rate, and other training parameters are the same as for the Set-Sequence model. The hyperparameters used are aligned with baseline hyperparameters from the corresponding papers.

Table 7: Hyper-parameter settings for each task.

| Hyper-parameter | Synthetic | Mortgage | Equity |
|---|---|---|---|
| Set-Seq layers | 5 | 6 | 6 |
| Sequence layers | 1 | 0 | 0 |
| $d_{\text{model}}$ | 800 | 300 | 64 |
| Learning rate | 0.003 | 0.003 | 0.003 |
| Dropout | 0 | 0.10 | 0.10 |
| Epochs | 40 | 40 | 30 |
| # Samples | 250 | — | — |
| Time steps | 100 | 50 | 246 |
| Number of units | 1–1000 | 2500 | 500 |
| Sequence model | LongConv | LongConv | LongConv |
| Chunk size $L$ | 3 | 3 | 3 |
| Set summary dim | 2 | 2 | 2 |
| $\phi$ output dim | 5 | 5 | 5 |
| Conv. weight decay | 0 | 0.05 | 0.05 |

**Compute resources**   All the experiments in the paper were conducted on a Linux cluster with 5 NVIDIA RTX A6000 GPUs, each with 49140 MB memory, running on CUDA Version 12.5. The cluster has 256 AMD EPYC 7763 64-Core Processor CPUs. For the synthetic task, each model was trained for less than 1 hour on 1 GPU. For the equities task, each train run (one run per validation year and random seed) took around 30 minutes on a single GPU. The experiments for the mortgage risk prediction task ran for around 1 hour per model training on one GPU.

**Data Sources**   For the mortgage risk case study we use the Version 2.0 CoreLogic® Loan-Level Market Analytics dataset released on July 5, 2022. For the equities portfolio prediction task we use base data derived from Center for Research in Security Prices, LLC (CRSP).

Table 8: Hyperparameter settings for the five sequence model baselines. For all models we use six layers, and the hidden dimension for a given task is the same as for the Set-Sequence model. The learning rate schedule, optimizer, and other training parameters are the same as for the Set-Sequence model.

| LongConv | | S4 | | H3 | |
|---|---|---|---|---|---|
| Kernel Length | 30 | $d_{\text{state}}$ | 64 | $d_{\text{state}}$ | 64 |
| Dropout | 0.0 | Learn $\theta, a$ | True | Head Dim | 1 |
| Nr Layers w/ Set | 6 | Skip Connection | True | Mode | diag |
| Set Embed Dim | 5 | Learning Rate | 0.001 | Measure | diag-lin |
| | | | | Learning Rate | 0.001 |
| Hyena | | Transformer | | | |
| Order | 2 | Heads | 8 | | |
| Filter Order | 64 | Causal | True | | |
| Num Heads | 1 | Learning Rate | 0.001 | | |
| Dropout | 0.0 | Dropout | 0.0 | | |

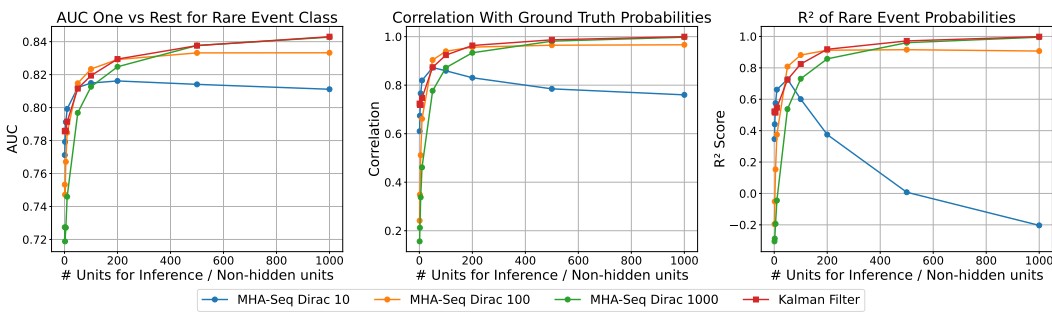

Figure 4: The performance for the MHA-Seq model depending on the training sampling method. $\delta_k$ denotes that all samples have input k units. The method are compute adjusted so that the number of epochs are scaled up to 400 (vs 40) for the case with input size 100, and scaled up to 4000 for input size 10.

# E  SYNTHETIC TASK: ADDITIONAL DETAILS AND RESULTS

## E.1  ABLATIONS ON THE TRAINING INPUT UNITS SAMPLING

As an ablation we consider sampling the number of units per batch with $D \sim \delta_k$, i.e. the number of input units is always k. The results are shown in Figure 4, where we use, 10, 100, 1000 samples (out of a total of 1000 units), compute adjusted, meaning that we train for 100 times more epochs (as each sample has 100 times fewer units) in the $n = 10$ case compared with $n = 1000$ observed units. We see a significant deterioration in performance when we only sample with 10 units, indicating that this is not sufficient to learn the joint structure in the data. The trade-off is that we get slightly better performance when we do inference on the same number of units that we trained on, but the generalization across units is significantly worse. We see that these methods provide worse generalization performance compared with the $(1 - \gamma)\delta_{1000} + \gamma \text{Logunif}$ sampling. We choose the LogUniform distribution as this gives roughly the same amount of samples in $(2, 4), (4, 8), \ldots (500, 1000)$, each can be viewed as a different data richness regime.

## E.2  INTERPRETABILITY

In Figure 5 we show an example of the set summary for several different number of observed units for one sample in the test set. As expected, the correlation increases as we observe more units, with the sample having $98\%$ correlation between the set summary in layer five with the true joint effect $\lambda_{0,t}$. Also note how the set summary in layer 5 learns a more smooth representation compared with layer 3.

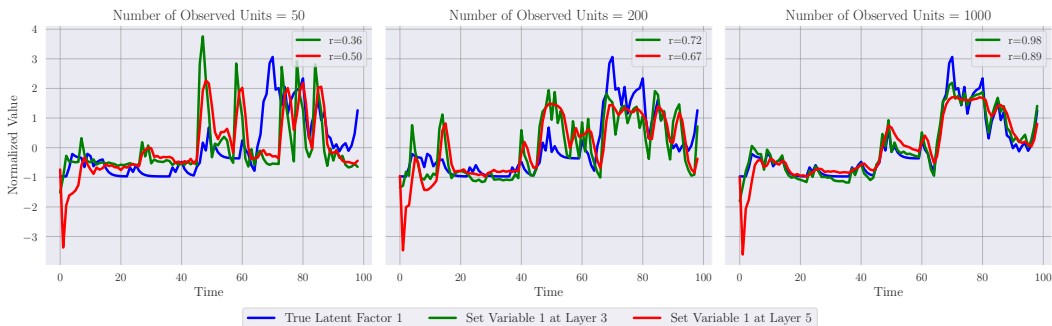

Figure 5: On the synthetic task with 1000 units in total, the Set-Sequence model's set summary in layer 3 and layer 5, learns the true joint effect $\lambda_{0,t}$, showing the interpretability of the Set-Sequence model. We see that as the number of observed units approach the fully observed case (1000 units), the set summaries become more aligned with the true effect, with a high correlation ($r = 0.98$) between the true joint effect and the set summary in layer five.

### E.3 KALMAN FILTER BASELINE

The synthetic task is described in Section 5. We assume that the transition matrix is known, and we also assume that the dynamics of $\lambda_t$, $\lambda_{t+1} = \beta\lambda_t + \alpha N_t$ are known, with the only unknown factor being not observing the true fraction of rare events $N_t$, but we only observe $\bar{N}_t$, where $\bar{N}_t$ is the observed fraction of rare events, only having access to n out of M units. The problem is now to infer the best estimate of $\lambda_t$ based on a partial observation of the fraction of rare events. We utilize a Kalman Filter Kalman (1960) to estimate $\lambda_t$ with $\hat{\lambda}_t$. The resulting filter estimates

$$\hat{\lambda}_{t+1} = \beta\hat{\lambda}_t + \alpha E[N_t](1 - K(t)) + \alpha K(t)\bar{N}_t,$$

a convex combination of the expected fraction of transitions to the absorbing state and the observed fraction of transitions to the absorbing state, where the Kalman Gain $K(t)$, is reflects how much noisily $\bar{N}_t$ estimates $N_t$. $K(t) = 1$ indicates that we fully observe all units, whereas $K(t) = 0$ correspond to the setting where none of the units are observed. The Kalman based baseline then uses $\hat{\lambda}_t$ to compute the estimated transition matrix 5, giving the predicted distribution of states in the next period. The Kalman Filter baseline assumes the true dynamics are known, whereas our model needs to learn the dynamics from a finite data. This provides an approximate upper bound for data-driven models.

**Approximate Dynamical Model**  We utilize that $N_t$ is a sum of $M$ Bernoulli random variables, and can hence be approximated with a normal distribution, where

$$N_t \sim \mathcal{N}(p_t, \sigma_{N_t}^2),$$
$$\sigma_{N_t}^2 = \frac{p_t(1 - p_t)}{M},$$
$$p_t = \frac{\lambda_t + \mu}{2 + x + (\lambda_t + \mu)(1 + 0.1x)}(1 + 0.1x),$$

Here, $p_t$ is the rare event transition probability, which is the expected fraction of rare events. We don't observe $N_t$ but rather $\bar{N}_t$, which approximately follow, by the Central Limit Theorem:

$$\bar{N}_t = N_t + \epsilon_t,$$

where

$$\epsilon_t \sim \mathcal{N}(0, \sigma_{\epsilon_t}^2),$$
$$\sigma_{\epsilon_t}^2 = \alpha^2 p_t(1 - p_t)\left(\frac{M - n}{M^2} + \frac{(M - n)^2}{M^2(n + 1)}\right).$$

**Kalman Filter**   We aim to estimate the parameter $\lambda_{t+1}$ using a Kalman Filter, where there are $M$ units in total and $n$ are observed. The high-level intuition is that to update $\lambda$, we want to use our current estimate of $\lambda$ and a convex combination of the expected value for $N_t$ and the partially observed value of $N_t$, where we put more weight on the observed value the higher n, the number of observed units, takes. Let $P_t$ be the variance of the estimate for $\lambda_t$. The Kalman update equations are then given by the prediction step

$$\hat{\lambda}_{t+1|t} = \beta\hat{\lambda}_t + p_t,$$
$$P_{t+1|t} = \beta^2 P_t + \sigma_{N_t}^2,$$

Kalman Gain computation

$$K_t = \frac{P_{t+1|t}}{P_{t+1|t} + \sigma_{\epsilon_t}^2},$$

and the update step

$$\hat{\lambda}_{t+1} = \hat{\lambda}_{t+1|t} + K_t\left(\bar{N}_t - p_t\right),$$
$$\hat{\lambda}_{t+1} = \max(\hat{\lambda}_{t+1}, 0), \quad \text{(to ensure positivity)}$$
$$P_{t+1} = (1 - K_t)P_{t+1|t}.$$

These equations provide a way to incorporate information from both the expected fraction of transitions to the absorbing state at time t, as well as the partially observed fraction of transitions to the absorbing state at time t.

**Visualizing the Kalman Filter**   Figures 6 show the Kalman Filter estimates of $\lambda_{x,t}$ for $x \in \{0, 1\}$, alongside the estimated gain $K(t)$ and the state estimate variance $P(t)$. In both cases, $n = 500$ corresponds to full observation (since 500 units have each feature value), resulting in zero estimation variance and $K(t) = 1$, as expected. As $n$ decreases, estimation variance increases and the Kalman gain decreases, causing the estimate to rely more on the prior mean than the noisy observations. This degrades tracking accuracy, as seen in the right panels. The transition model uses $\mu = 0.001$, $\alpha = 4$, and $\beta = 0.5$.

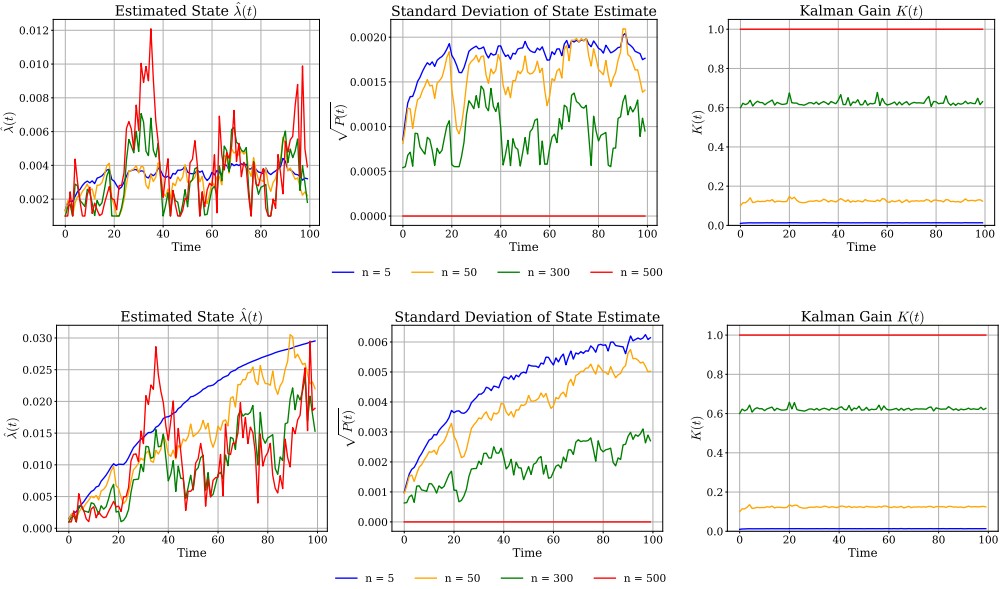

Figure 6: Kalman Filter estimates of $\lambda_{x,t}$, state estimate variance, and Kalman gain $K(t)$ for $x = 1$ (top) and $x = 0$ (bottom). Both cases use 500 total units per feature value. As $n$ decreases, estimation variance increases and the Kalman gain $K(t)$ drops, causing less accurate tracking of $\lambda_{x,t}$. The transition model uses $\mu = 0.001$, $\alpha = 4$, and $\beta = 0.5$.

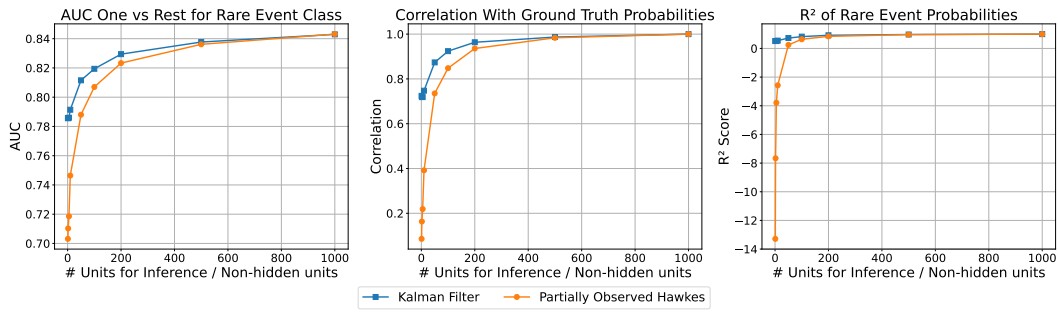

Figure 7: The Kalman Filter vs "Partially Observed Hawkes" which is the Kalman model where $K(t)$ is fixed equal to 1.

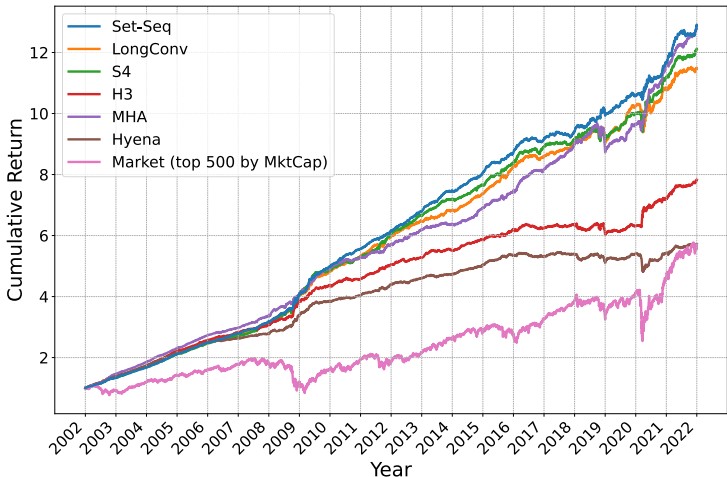

Figure 8: Cumulative returns from 2002 to 2021 for the Set-Sequence model and sequence model baselines. We also show the equally weighted market portfolio of the largest 500 stocks by market cap.

**Improvement of Kalman Filter over simple estimate** Here we compare the Kalman Filter with other simple baseline models. Figure 7 shows the performance of the Kalman vs the baseline that sets a fixed gain $K(t) = 1$, meaning that it estimates $\lambda_t$ with $\tilde{\lambda}_t$, where

$\tilde{\lambda}_{t+1} = \beta \tilde{\lambda}_t + \alpha \bar{N}_t$, i.e. it uses the observed fraction of transitions to the absorbing state to estimate $\lambda_t$ and does not consider the expected number of default events.

## F  EQUITY PORTFOLIO PREDICTION: ADDITIONAL DETAILS AND RESULTS

Figure 8 shows the cumulative returns for the Set-Sequence model compared with the baselines.

**Transaction Cost Analysis** We test the Set-Sequence model and the other baseline models under realistic transaction costs. In particular, for each day we subtract the transaction costs from the daily portfolio returns, which are used to compute the Sharpe ratio. Following Guijarro-Ordonez et al. (2022), we use the following model for the transaction cost for portfolio weights $w_t$ at time t, and $w_{t-1}$ at t-1:

$$\text{cost}(w_t, w_{t-1}) = 0.0005 \|w_t - w_{t-1}\|_1$$

$$+ 0.0001 \|\max(-w_t, 0)\|_1.$$

The return at time t, can then be expressed as $r_{net,t} = r_t - \text{cost}(w_t, w_{t-1})$. We now train the Set-Sequence model and the Sequence model baselines with a modified objective using the net return

Table 9: Out of sample annualized performance metrics net transaction costs 2002 - 2016, trained with a net sharpe ratio objective. Details of the models other than the Set-Sequence model are found in Guijarro-Ordonez et al. (2022). The training setting for the sequence models and Set-Sequence model is in Appendix D.

| Model | Net Sharpe | $\mu$ % | $\sigma$ % |
|---|---|---|---|
| CNN+Trans K=0 (IPCA) | 0.52 | 8.5 | 16.3 |
| CNN+Trans K=1 (IPCA) | 0.85 | 5.9 | 6.9 |
| CNN+Trans K=3 (IPCA) | 1.24 | 6.6 | 5.4 |
| CNN+Trans K=5 (IPCA) | 1.11 | 5.5 | 5.0 |
| CNN+Trans K=10 (IPCA) | 0.98 | 5.1 | 5.2 |
| CNN+Trans K=15 (IPCA) | 0.94 | 4.8 | 5.1 |
| LongConv | 1.64 | 7.41 | 4.52 |
| S4 | 2.17 | 8.31 | 3.82 |
| H3 | 0.76 | 3.71 | 4.89 |
| Transformer | 1.59 | 7.40 | 4.66 |
| Hyena | 0.60 | 3.59 | 6.00 |
| Set-Seq Model (ours) | **2.46** | 8.4 | 3.4 |

Sharpe as the loss function. In Table 9 we show the net Sharpe ratio after costs in Jan. 2002 - Dec. 2016. We see the gain from modeling the cross-section as the Set-Sequence model outperforms all the sequence models in terms of net Sharpe Ratio, as well as the domain specific CNN-Transformer.

**Equity Task Features** Table 10 shows the rank normalized features. In addition to a subset of the features in Chen et al. (2024) we include features for daily return, weekly return, weekly volatility, daily volume. In order to keep the level information, we also include cross-sectional median features in Table 11, following Chen et al. (2024).

# G  MORTGAGE RISK PREDICTION: ADDITIONAL DETAILS AND RESULTS

## G.1  ADDITIONAL DATA DESCRIPTIONS

We provide additional details about the CoreLogic Loan-Level Market Analytics Dataset. The dataset is over 870 GB and contains month-by-month transition data for mortgages in the US. The dataset starts in 1988, and has data to 2024, with data from 30000 ZIP codes in the US. We restrict ourselves to the top 4 ZIP codes, in terms of number of active loans, in the greater Los Angeles area and leave to follow up work to train on the full dataset. The ZIP Codes are: 92677, with 109642 loans in Orange County; 93065, with 98673 loans in Simi Valley; 91709, with 95497 loans in Chino Hills; 92336 with 94794 loans in Fontana. We follow the same data filtering procedure as in Sadhwani et al. (2020), and use a subset with 52 of the features included there, we filter out loans if any of the following features are not present: FICO score, Original balance, Initial interest rate, and Current State. Some other features we use include Original LTV, Unemployment Rate, Current Interest Rate, and National Mortgage Rate. We deal with missing data in other features with the missing indicator method, see, for example Little & Rubin (2014). After filtering $117,523$ loans remain, each active for on average 45 months, for a total of around 5 million loan month transitions. Figure 9 shows the empirical transition probabilities across the dataset, and Figure 10 shows the transition counts on the sampled test set. Figure 11 shows the number of active loans, the prepayment rate, and the foreclosure rate over time on the dataset. For example, note the elevated foreclosure rates during and after the 2008 financial crisis. Table 12 shows the full set of features for the Set-Sequence model.

## G.2  MODEL FITTING

We create an additional "non active" state that serves as a mask to ensure all loan sequences cover the full dataset. We sample loans in the following way in each batch: First, randomly pick a start time, then collect all active loans at that start time, and pick a subset N of them (set to 2500 in our experiments). We make this operation more efficient by first sorting the loans by origination date. Since the Set-Sequence model uses a context window of 50 time steps we sample using overlap

Table 10: Firm-specific characteristics (six categories) used as features in the equity-portfolio-optimization task. Construction details are in the Internet Appendix of Chen et al. (2024).

| **Past Returns** | | **Value** | |
|---|---|---|---|
| r2_1 | Short-term momentum | A2ME | Assets to market cap |
| r12_2 | Momentum | BEME | Book-to-market ratio |
| r12_7 | Intermediate momentum | C | Cash + short-term inv. / assets |
| r36_13 | Long-term momentum | CF | Free cash-flow / book value |
| ST_Rev | Short-term reversal | CF2P | Cash-flow / price |
| Ret_D1 | Daily return | Q | Tobin's Q |
| Ret_W1 | Weekly return | Lev | Leverage |
| STD_W1 | Weekly volatility | E2P | Earnings/Price |
| **Investment** | | **Trading Frictions** | |
| Investment | Investment | AT | Total assets |
| NOA | Net operating assets | LME | Size |
| DPI2A | Change in PP&E | LTurnover | Turnover |
| | | Rel2High | Closeness to 52-week high |
| | | Resid_Var | Residual variance |
| **Profitability** | | **Trading Frictions (cont.)** | |
| PROF | Profitability | Spread | Bid–ask spread |
| CTO | Capital turnover | SUV | Standard unexplained volume |
| FC2Y | Fixed costs / sales | Variance | Variance |
| OP | Operating profitability | Vol | Weekly Trading volume |
| PM | Profit margin | RF | Risk-free rate |
| RNA | Ret. on net operating assets | Beta | Beta with market |
| D2A | Capital intensity | | |
| **Intangibles** | | | |
| OA | Operating accruals | | |
| OL | Operating leverage | | |
| PCM | Price-to-cost margin | | |

Table 11: Cross-sectional median firm-characteristic variables and their stationary transformations (*t*Code). Each feature is the cross-sectional median of the underlying firm characteristic. *Note:* The transformations (*t*Code) are (1) no transformation; (2) $\Delta x_t$; (3) $\Delta \log(x_t)$; (4) $\Delta^2 \log(x_t)$.

| **Variable** | *t***Code** | **Variable** | *t***Code** | **Variable** | *t***Code** | **Variable** | *t***Code** |
|---|---|---|---|---|---|---|---|
| A2ME | 3 | AT | 4 | BEME | 3 | Beta | 1 |
| C | 3 | CF | 2 | CF2P | 3 | CTO | 3 |
| D2A | 3 | DP2A | 3 | E2P | 3 | FC2Y | 3 |
| Investment | 3 | Lev | 3 | LME | 4 | LTurnover | 3 |
| NOA | 3 | OA | 2 | OL | 3 | OP | 3 |
| PCM | 3 | PM | 3 | PROF | 3 | Q | 3 |
| Ret_D1 | 2 | Rel2High | 3 | Resid_Var | 3 | RNA | 3 |
| r2_1 | 2 | r12_2 | 2 | r12_7 | 2 | r36_13 | 2 |
| Spread | 3 | ST_REV | 2 | SUV | 1 | Variance | 3 |
| Vol | 4 | STD_W1 | 3 | Ret_W1 | 2 | | |

Table 12: Feature definitions and possible values. We require that current state, FICO score, Original balance, and Initial Interest Rate are available, and use a missing indicator when other features are unavailable. The feature vector including the missing indicators has dimension 52.

| Feature | Values |
|---|---|
| Current state | Current, 30 Days Delinquent, 60 Days Delinquent, 90+ Days Delinquent, Paid Off, Foreclosure, Real-Estate Owned |
| FICO score | Continuous |
| Original balance | Continuous |
| Initial interest rate | Continuous |
| Original LTV | Continuous |
| State unemployment rate | Continuous |
| National mortgage rate | Continuous |
| Current balance | Continuous |
| Current interest rate | Continuous |
| Scheduled monthly principal and interest | Continuous |
| Scheduled principal | Continuous |
| Days delinquent | Continuous |
| Prime mortgage flag | True, False |
| Convertible flag | True, False |
| Pool insurance flag | True, False |
| Insurance only flag | True, False |
| Prepay penalty flag | True, False |
| Negative amortization flag | True, False |
| Time since origination | $< 1$ Year, $< 5$ Years, $< 10$ Years |
| Original term | $< 17$ Years |
| Number of 30-day delinquencies (past 12 months) | 0–12 |
| Number of 60-day delinquencies (past 12 months) | 0–12 |
| Number of 90+-day delinquencies (past 12 months) | 0–12 |
| Number of 30-day foreclosures (past 12 months) | 0–12 |
| Number of Current occurrences (past 12 months) | 0–12 |

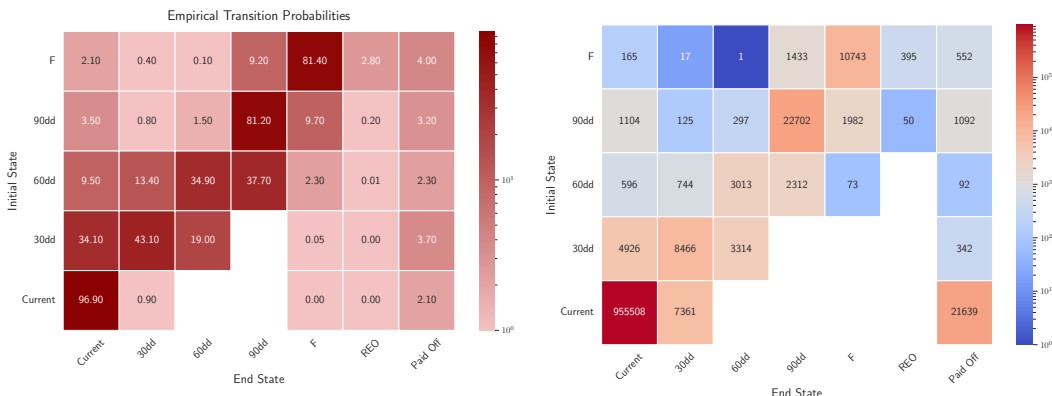

Figure 9: Empirical Transition Probabilities for the full top 4 ZIP codes on the combined train, validation and test dataset.

Figure 10: Top 4 ZIP transition counts on the test set January 2010 - December 2023.

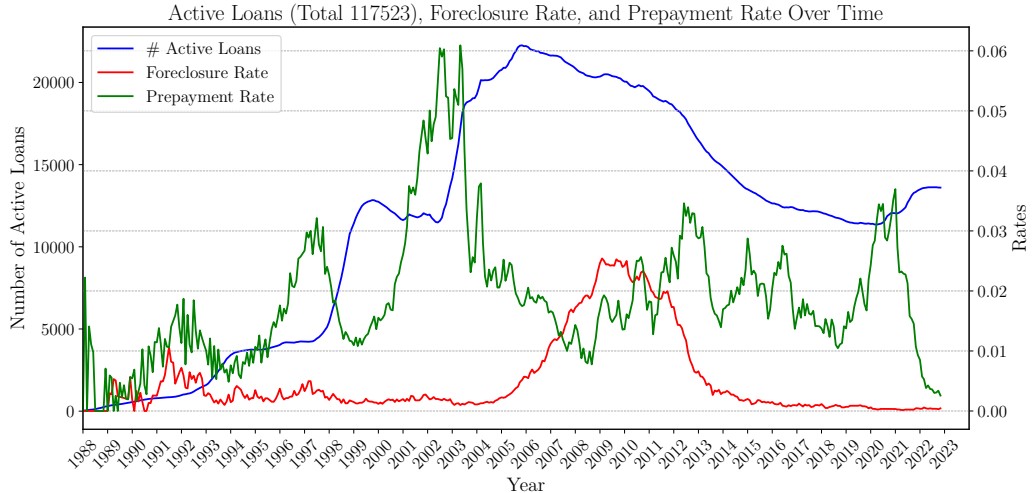

Figure 11: Active Loans, Foreclosure Rate, and Prepayment Rate over time.

between the train, validation, and test sets and only compute train gradients and performance metrics on the masked indices within the respective partition.

### G.3 Additional Experiments Mortgage Risk Prediction

#### G.3.1 Interpretability

In Figure 12 in the Appendix, the foreclosure rate, a known source of cross unit dependency, is shown to have a Pearson correlation of 0.67 with the set summary in the first Set-Sequence layer, indicating the interpretability of the learned set summaries.

#### G.3.2 Transition Analysis

Figure 13 shows the one vs rest AUC conditioned on the initial state for the Set-Sequence model, averaged over 10 seeds of samples on the test set, where for each seed we sample 25 sequences of 2500 loans from the test set.

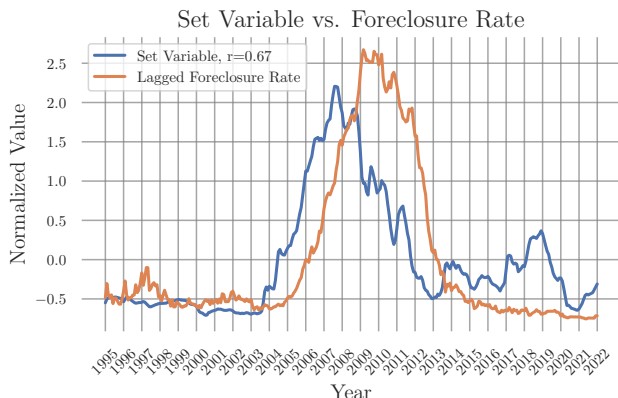

Figure 12: Foreclosure rate over the dataset, as well as the learned set representation in the first set layer in the neural network. The Set-Sequence model learns to predict the foreclosure rate for joint default modeling.

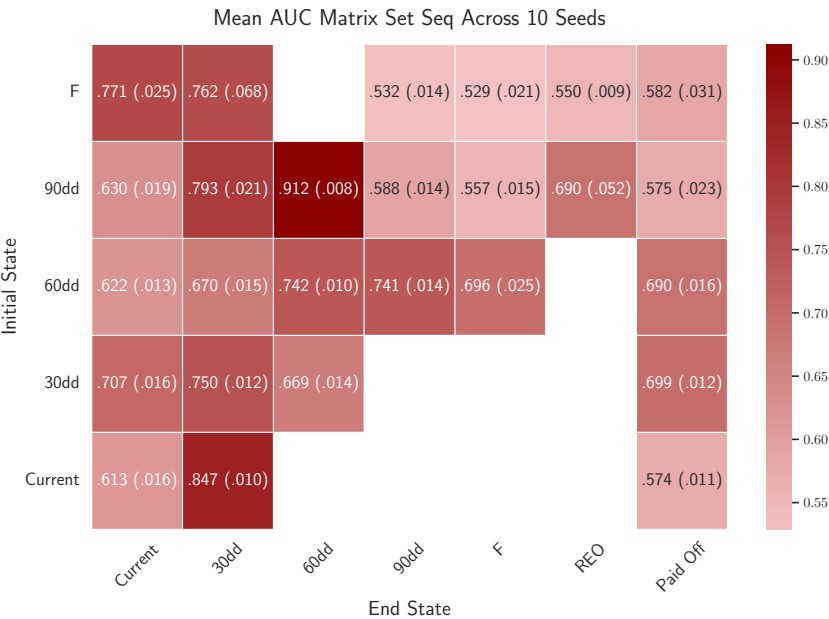

Figure 13: AUC Matrix for the Set-Sequence Model with 50 Features, averaged across 10 random seeds. In parenthesis we show the standard deviation of the AUC for that transition. We only show the AUC for transitions that happened at least 10 times for each seed.

### G.3.3 YEAR-BY-YEAR RESULTS

In this subsection, we aim to understand the robustness of the Set-Sequence model across time for the mortgage risk task. To that end, we first describe our refitting approach with time.

**Yearly Refitting** To emphasize recent data, we weight training windows with an exponential–decay schedule whose half-life is $\tau$ (e.g., 24 months). Define

$$\alpha = \frac{\ln 2}{\tau}, \qquad t_{\max} = \max_i t_i.$$

Each window at time $t_i$ receives the (unnormalized) weight

$$w_i = \exp\big[\alpha\,(t_i - t_{\max})\big].$$

Consequently, a window that is $\tau$ time units older than $t_{\max}$ has half the weight of the most recent window. During refitting we sample windows with probability proportional to $w_i$, biasing the training set toward more recent periods.

Figure 14 shows the improvement for the Set-Sequence model when using refitting method with $\tau = 24$ months compared with not retraining and not weighting the retraining samples. We see a performance improvement with the weighted refitting.

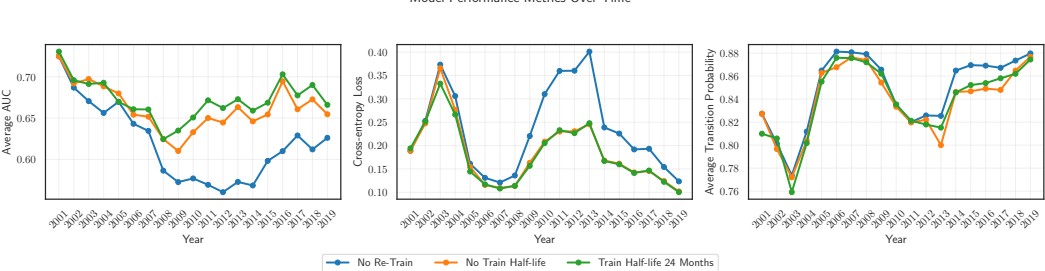

Figure 14: Comparing three methods for data refitting: 1. not retraining 2. retraining with equal weight on all times, and 3. retraining with more weight on later times.

**Results**  We compare the Set-Sequence model, the Gated Selection model, and the models from Sadhwani et al. (2020) in the setting where we do a base training for 15 epochs up until year 2002, and then for each epoch the train-set gets extended one year further. We save a checkpoint each year the model is extended.

In addition we up-weight the more recent train data, this is described in Section G.3.3. Figure 15 and Figure 16 shows the results, comparing the Set-Sequence, Gated Selection (see Appendix C), NN, and Logistic model, all using the same retraining method, and where we evaluate on one year at a time. We see that the challenging years around the financial crisis affects the logistic model the most.

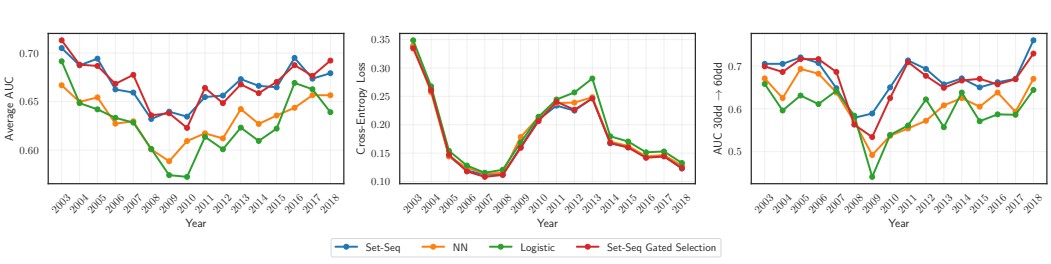

Figure 15: Average AUC, Cross-Entropy, and 30dd-60dd AUC for the Set-Sequence, NN, and Logistic model, all using the same retraining method, and where we evaluate on one year at a time.

However, the relative performance of the Set-Sequence model and the NN model remain broadly the same, although it widens somewhat during 2009.

### G.3.4    INTERPRETABILITY

We show that the Gated Selection model is interpretable. To see the patterns learned in the gating matrix $G$ more clearly we first sort the rows/columns by the zip-code (white lines), then, within each ZIP-code, we sort by prime/subprime/unknown loan type (red lines). From Figure 17 we see that the selection matrix G learn to distinguish between these categories, which is aligned with the importance of these categorizations suggested in prior work. This means that the set-summary of a mortgage that, say is in ZIP code 1 and is subprime, will be a weighted sum, with the majority of the weight on other subprime mortgages in ZIP code 1. Another point to note here is that the gating matrix meaningfully change with economic cycles, so by considering the year 2018 instead of 2012

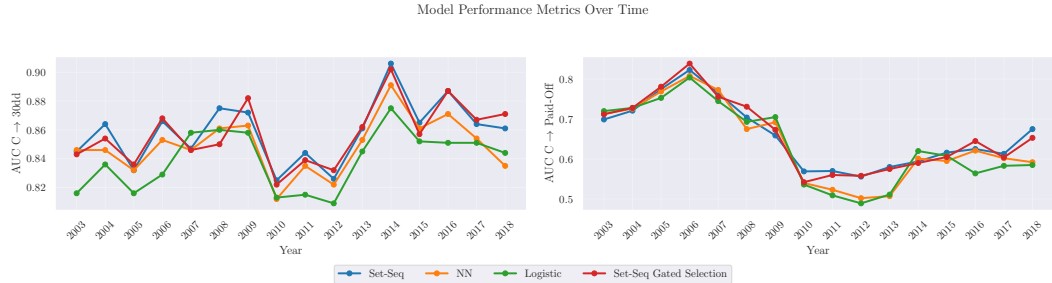

Figure 16: The AUC per year for different transitions.

gives a different structure (highlighting for example different economic conditions evolving through time in the different ZIP-codes).

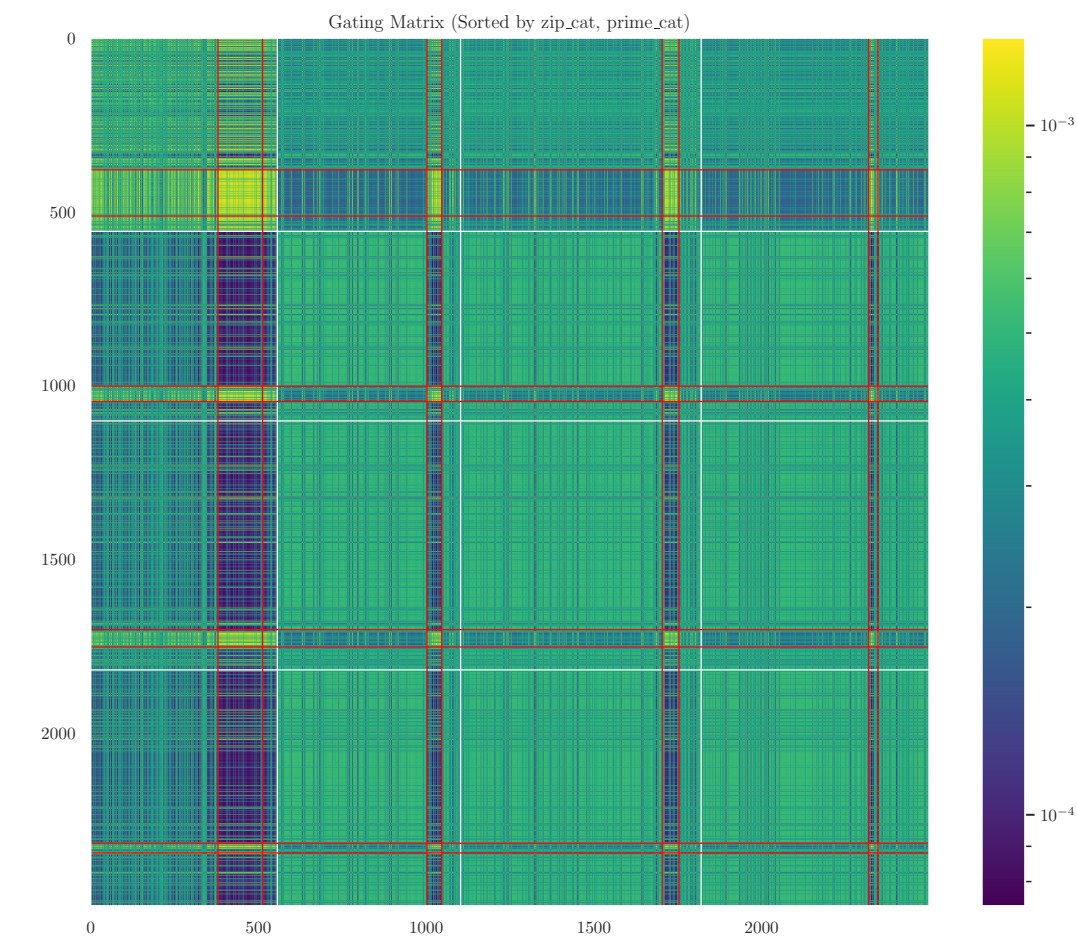

Figure 17: The selection matrix $G$, sorted by ZIP-code (white lines), and prime/subprime/unknown status (red lines).

# H  ADDITIONAL METRIC DETAILS

Here we present formal definitions of metrics used in the paper. We are interested in considering these metrics for the class k corresponding to the absorbing (rare) class. Given access to true transition

probabilities in our synthetic task framework, we establish test set evaluation metrics focused on the absorbing (rare) state. For this absorbing state $k$, we define the area under the receiver operating characteristic curve (AUC) metric as

$$\mathrm{AUC}(k) = P\big(y_{1,k} > y_{2,k} \mid y_1 = k,\ y_2 \neq k\big)$$
$$+ \frac{1}{2}\, P\big(y_{1,k} = y_{2,k} \mid y_1 = k,\ y_2 \neq k\big).$$

which characterizes the model's discriminative capacity in identifying transitions to the absorbing state. By adding negligible noise to the predictions, all scores will be different and the second term will equal zero. Given a dataset of $n$ samples, $P$ of which belongs to class $K$ and $N$ that does not belong to class $K$, the AUC can be computed with

$$AUC(k) = \frac{1}{NP} \sum_{i=1, j=1}^{n,n} 1_{\hat{y}_{i,k} > \hat{y}_{j,k}} 1_{y_i = k} 1_{y_j \neq k}.$$

This is more efficient than creating the ROC curve and computing the area underneath it. To quantify prediction accuracy for the absorbing state, we compute the correlation between predicted and true transition probabilities

$$\mathrm{Corr}(p_k, \hat{p}_k) = \frac{\mathrm{Cov}(p_k, \hat{p}_k)}{\sqrt{\mathrm{Var}(p_k) \cdot \mathrm{Var}(\hat{p}_k)}}.$$

The coefficient of determination $R^2$ measures the proportion of variance captured by predictions for transitions to the absorbing state

$$R^2 = 1 - \frac{\sum_{i=1}^{n}(p_{i,k} - \hat{p}_{i,k})^2}{\sum_{i=1}^{n}(p_{i,k} - \bar{p}_k)^2},$$

where $\bar{p}_k = \frac{1}{n}\sum_{i=1}^{n} p_{i,k}$. Our synthetic setup enables direct probability-based evaluation, rather than relying on predicted class labels. The Kullback-Leibler divergence measures the difference between the predicted and true probability distributions

$D_{KL}(p_k | \hat{p}_k) = \frac{1}{n}\sum_{i=1}^{n} p_{i,k} \log\left(\frac{p_{i,k}}{\hat{p}i,k}\right)$. This metric is non-negative and equals zero if and only if $p_{i,k} = \hat{p}_{i,k}$ for all $i$. Unlike cross-entropy between the true labels and the predicted probabilities, the KL divergence between the predicted and true transition probabilities explicitly shows the deviation from a perfect model.

