# OpenReview forum: "A Set-Sequence Model for Time Series"
_ICLR.cc/2026/Conference — Submitted to ICLR 2026_

### Official Review · Reviewer_1etJ · 2025-10-29

**Soundness:** 2
**Presentation:** 2
**Contribution:** 1
**Rating:** 2
**Confidence:** 5

**Summary:**

The paper proposes Set-Sequence, a model for large-scale time series prediction. It first uses a Set module to extract a permutation-invariant population summary, then combines it with individual features in a Sequence module (e.g., Transformer or RNN) for prediction. Experiments show improved accuracy, efficiency, and interpretability across multiple tasks.

**Strengths:**

The paper presents a clear and modular architecture that integrates a permutation-invariant Set module with standard sequence models, offering linear scalability and theoretical grounding through connections to Deep Sets. Its empirical evaluation—spanning large-scale financial datasets such as CRSP equities and CoreLogic mortgages—demonstrates strong predictive and economic performance. However, the validation remains limited to financial domains, and the interpretability analysis, while suggestive of meaningful latent structure, is primarily correlational and not deeply explored.

**Weaknesses:**

1) The motivation is partially unclear: while the paper aims to handle large cross-sections of time series efficiently, it is not well distinguished from the classical setting of multivariate time series modeling. If each entity were viewed as a “variable,” the proposed Set-Sequence formulation would resemble feature aggregation rather than feature selection. However, the paper does not clarify this distinction nor position its contribution relative to multivariate temporal modeling, leaving its conceptual novelty somewhat ambiguous.

2) While the paper motivates itself by the need to capture cross-entity interactions, it does not explicitly model pairwise or structured dependencies as in graph or attention-based methods. Instead, it addresses a different, more specific challenge: efficiently summarizing population-level context under exchangeability when the number of entities is extremely large. This makes the contribution one of scalability and global aggregation, rather than relational modeling in the strict sense.

3) The dataset includes only the top 500 market-cap equities per period and excludes missing returns, which may introduce survivorship or selection bias. It is recommended to include a less-filtered baseline (e.g., fixed universe or delisted stocks) and year-by-year sensitivity analyses to ensure robustness.

4)  The model assumes exchangeability and uses pooled moments, which may fail under strong non-exchangeable or localized dependencies. A small synthetic experiment comparing Set-Seq with attention or graph models under such conditions would clarify its expressive limits.

5) The paper demonstrates the expressive capability of the model; however, in practical testing, its performance shows a clear gap compared to attention-based mechanisms. The paper does not clearly explain the cause of this discrepancy and only highlights the model’s superiority in terms of time complexity.

6)  The paper mainly uses mean and polynomial pooling. Including or emphasizing ablations on alternative pooling methods (e.g., quantile, robust, or group-wise pooling) would better illustrate expressivity–cost trade-offs.

**Questions:**

1) How is the proposed setting fundamentally different from classical multivariate time series modeling (multivariate time series feature selection, reweighting), beyond computational efficiency?

2) If the model does not explicitly capture pairwise or structured dependencies, can it truly model cross-entity interactions?

3) Does restricting the dataset to the top 500 equities introduce survivorship or selection bias, and how robust are the results under a less-filtered universe?

4) How does the exchangeability assumption affect performance when dependencies are localized or non-exchangeable?

5) Why does the model underperform attention-based baselines despite its claimed expressivity advantages?

Would exploring alternative pooling functions (e.g., quantile or group-wise) change the expressivity–efficiency trade-off?

---

> ### Author Response · Authors · 2025-11-27
> **Rebuttal by Authors**
>
> Thank you for your review, in particular the concern around our setting vs multivariate time series.
>
> #### W0/Q0. The motivation is partially unclear: while the paper aims to handle large cross-sections of time series efficiently, it is not well distinguished from the classical setting of multivariate time series modeling.
>
> This distinction is both significant and central to our setting, as summarized below. We welcome feedback on how to make this clearer in the paper.
>
> **Classical Multivariate Time Series (MTS):** A standard MTS problem consists of a *single* multivariate time series $(U, y)$, where $U = [x_1, x_2, .., x_T]$ is the time-series input and $y$ is the prediction target. Such a sequence $U$ is exactly what we call a *unit* in our paper. For example, $U_{\text{GOOG}}$ denotes the time series of Google's features. Classical MTS models operate on this single unit and therefore cannot capture cross-sectional relationships between multiple units.
>
> **Our Set–Sequence Setting:** In our case, one problem instance consists of a *set* of such units {$(U_1, y_1), \ldots, (U_N, y_N)$}, each with its own multivariate time series. This structure allows the model to harness cross-sectional information that is not present in any single unit. For example, a basket of $N=3$ stocks such as $(U_{\text{GOOG}}, U_{\text{NVDA}}, U_{\text{AAPL}})$ may exhibit shared industry-wide trends that no individual sequence contains. Crucially, the number of units $N$ varies across instances, so we cannot represent the input as a fixed $N \times d$ multivariate time series as in classical MTS.
>
> #### Q1. The model assumes exchangeability and uses pooled moments, which may fail under strong non-exchangeable or localized dependencies. A small synthetic experiment comparing Set-Seq with attention or graph models under such conditions would clarify its expressive limits.
>
> We clarify the validity of exchangeability in our common response at the top. Our suite of synthetic experiments include the scenario of localized dependencies: a subset of units belongs to class 1 and another subset to class 2 (analogous to prime vs subprime loans, each class has a distinct dynamical structure). The class is provided as a unit-level feature. In our experimental setup, we find that the Set-Sequence model with a set-summary of dimension two learns the default rate of class 1 units in the first set variable and the default rate of class 2 units in the second set variable, illustrating both expressivity and interpretability of simple mean aggregation.
>
> #### Q2. If the model does not explicitly capture pairwise or structured dependencies, can it truly model cross-entity interactions?
>
> Our model can capture pairwise interactions if the interaction is present as a feature in the model. The richer the features, the more expressive the Set–Sequence architecture becomes.
>
> #### Q3. Does restricting the dataset to the top 500 equities introduce survivorship or selection bias, and how robust are the results under a less-filtered universe?
>
> We follow the same data methodology as in prior published work to ensure a clear comparison. We use the top 500 equities since these are the most liquid ones, with the lowest transaction costs (making our transaction cost analysis realistic), and since these are harder to predict compared with small cap stocks. These will also have much less price impact when buying/selling. This set is also often targeted by institutional investors.
> The most important result is that adding the cross sectional set modeling improves over baseline trained on the same data, just using the sequence models on with a large margin.
>
> #### Q4. How does the exchangeability assumption affect performance when dependencies are localized or non-exchangeable?
>
> As several reviewers asked similar questions, we provide one unified answer in the common response at the top.
>
> #### Q5. Why does the model underperform attention-based baselines despite its claimed expressivity advantages?
>
> The attention based set aggregation method MHA-Seq is a contribution of our work rather than a baseline. Prior attention based methods that have used attention have viewed each feature as one token in the cross section, however, in our work, we view each unit (with multiple features per timestep) as a token, so that units are compared rather than features. The MHA-Seq, is more expressive in that it can handle non-exchangeable data but at substantially higher computational cost.
>
> #### Q6. The paper mainly uses mean and polynomial pooling. Including or emphasizing ablations on alternative pooling methods (e.g., quantile, robust, or group-wise pooling) would better illustrate expressivity–cost trade-offs.
>
> We already evaluate three cross-sectional methods: mean pooling, MHA-Seq, and the gated-selection mechanism (a weighted mean). Additional experiments with gated selection are provided in Appendix C.

---

### Official Review · Reviewer_eEp8 · 2025-10-30

**Soundness:** 3
**Presentation:** 3
**Contribution:** 2
**Rating:** 4
**Confidence:** 5

**Summary:**

This paper addresses high-dimensional time series prediction with correlated units (e.g., stocks, loans, sensors). It needs to capture cross-sectional dependencies among M units and temporal dynamics over T steps, but direct joint modeling is computationally infeasible. Existing methods rely on hand-crafted cross-sectional features, fail to adapt to dynamic unit counts, and miss latent correlations—though the problem itself is important.

The paper proposes the Set-Sequence model (Set module for linear-complexity permutation-invariant cross-sectional summaries; Sequence module for unit temporal dynamics). Evaluated on a synthetic loan default task, 36,600 U.S. equities, and 5M mortgage records, it outperforms baselines (Transformer, S4, domain models): 4.4–10.2× lower KL divergence, 22%–42% higher equity Sharpe ratio, and 0.683 mortgage AUC.

**Strengths:**

Strength 1: Set-Sequence models cross-sectional dependencies as a permutation-invariant set summary. Linear pooling plus a shallow readout is proven to uniformly approximate any continuous permutation-invariant target. This result supplies an expressive lower bound under exchangeability and directly informs the choice of pooling order and hyper-parameters.

Strength 2: The module plugs into any temporal backbone with Θ(M) complexity. A single scan and per-unit temporal update reduce FLOPs by an order of magnitude relative to self-attention when M≥2500. Units can be added or dropped at inference without retraining, and memory grows linearly, so one A6000 GPU handles 5 M samples.

Strength 3: Consistent, measurable gains are observed on synthetic contagion, equity-portfolio, and mortgage-default tasks. Compared to strong baselines, it achieves 10× KL reduction, +42 % Sharpe, and +4 average AUC, with improvements that grow monotonically with pool size. The learned summary correlates 0.67 with the realized foreclosure rate, delivering an auditable macro signal for risk management.

**Weaknesses:**

Limited methodological novelty: The proposed Set-Sequence model lacks substantial innovation. It simply combines an existing permutation-invariant set encoder with a standard sequential backbone, following the design principles of Deep Sets and Set Transformer with only minor modifications. Theoretical results are mostly restatements of known approximation properties rather than new insights. Overall, the contribution feels more like an engineering integration than a novel modeling paradigm.

Outdated and insufficient experimental comparisons: The experiments rely on relatively old baselines and omit recent state-of-the-art time-series architectures such as iTransformer, TimeMixer, PatchTST, and Mamba. The reported improvements over older models therefore do not convincingly demonstrate the superiority of the proposed method. Moreover, the empirical evaluation lacks comprehensive analyses such as cross-task generalization, robustness, or hyperparameter sensitivity.

Shallow theoretical and architectural analysis: The claimed linear scalability is only valid under a simple mean-pooling assumption, which is not evaluated in heterogeneous or strongly correlated scenarios. The paper does not analyze how this simplification affects representational capacity or generalization performance, nor does it study how the look-back parameter interacts with temporal dependencies. Consequently, the theoretical analysis remains shallow and fails to substantiate the claimed advantages of the architecture.

**Questions:**

Sensitivity to look-back parameter 𝐿: The paper fixes 𝐿=3 without justification. How sensitive is the model to this choice? Does increasing
𝐿 enhance long-term dependency modeling or simply increase computation? A short sensitivity analysis would clarify its role.

Robustness of linear aggregation: The Set module employs mean pooling for cross-sectional summarization. Given potential heterogeneity among units, is this pooling stable or biased toward large or noisy samples? Have the authors explored normalization or adaptive weighting schemes?

---

> ### Author Response · Authors · 2025-11-27
> **Rebuttal by Authors**
>
> Thank you for your review and thoughtful suggestions. We answer your questions below.
>
>
> #### W1. Limited methodological novelty: The proposed Set-Sequence model lacks substantial innovation. It simply combines an existing permutation-invariant set encoder with a standard sequential backbone, following the design principles of Deep Sets and Set Transformer with only minor modifications...
>
> Our contribution is not in the set mechanism itself, but rather in how it is applied: by applying the set operation on a set with multivariate time series, and augmenting the features with the resulting set-summary, a practical, important problem of joint modeling many units is tackled.  The model is simple, we don’t view this as a limitation but rather as a strength, as this makes it easy to understand each component of the model. In particular, it allows any multivariate time series model to work in the setting where we have a set of multiple units, with joint dependencies, each of which is a multivariate time series. Deep Sets, and Set Transformer are neither discussing time series modeling at all.
>
> #### W2. Outdated and insufficient experimental comparisons…
>
> While the mentioned architectures are recent, they have not been applied to the setting we are analyzing, hence it’s not clear that they are better than other approaches. For instance, mamba would be possible to use in the sequence part of the Set-Sequence model, hence it’s not a direct comparison but would rather be one further example of a Sequence model that works in our framework. We believe that the current wide range of sequence models we use make it clear that the Set-Sequence paradigm is able to use any Sequence model, rather than just the ones we show (from Conv, Transformer, SSM, and mixed paradigms).
>
>
> #### W3. Shallow theoretical and architectural analysis…
> In our framework, we show that if you use a quadratic pooling operation such as attention, while it still scales quadratically. However, it scales quadratically in the number of units rather than in the number of units times the number of features per unit, which has a large effect on downstream performance.
>
> #### Q1. Sensitivity to look-back parameter 𝐿: The paper fixes 𝐿=3 without justification. How sensitive is the model to this choice? Does increasing 𝐿 enhance long-term dependency modeling or simply increase computation? A short sensitivity analysis would clarify its role.
> Thank you for this excellent suggestion. We have run this experiment, with the results and a detailed answer in the common response, showing robustness to different values of L.
>
> #### Q2. Robustness of linear aggregation: The Set module employs mean pooling for cross-sectional summarization. Given potential heterogeneity among units, is this pooling stable or biased toward large or noisy samples? Have the authors explored normalization or adaptive weighting schemes?
>
> We do feature normalization so each feature should have mean 0 so it’s not likely that we would see features with very large weight. Yes, we have considered an adaptive weighting scheme explored in the appendix on the Gated Selection module, that based on the input computes a weight for each term in the aggregation, in Appendix C in our paper. We presented this method in the Appendix as it had similar empirical performance compared with the simpler linear aggregation on the task we consider.

---

### Official Review · Reviewer_VTYC · 2025-10-30

**Soundness:** 3
**Presentation:** 2
**Contribution:** 3
**Rating:** 6
**Confidence:** 4

**Summary:**

The paper tackles prediction over large cross-sections of units (loans, stocks, customers) where each unit is a time series. The authors propose a Set-Sequence architecture that factors the problem into two natural dimensions: Set modeling across entities at each time step (the “cross-section”). Sequence modeling through time for each entity (the “temporal” dimension). Experiments on (i) a synthetic contagion/default task, (ii) U.S. equity portfolio construction, and (iii) mortgage risk prediction (~5M loan-months; 52 features) show consistent gains, plus interpretable set summaries that correlate with latent factors.

**Strengths:**

-  The idea is clear: set over units → sequence over time.
- The method is scalable,  with theoretical complexity analysis, and can be extended when M is large.
- Empirical results are broad and strong, both in Synthetic contagion and real-world datasets.
- The model gives the Interpretability.

**Weaknesses:**

- The Set-Sequence architecture extends the idea of multiple instance learning mean pooling to the temporal domain. There are a few related works that are encouraged to be discussed or compared.

[1] Zaheer, Manzil, et al. "Deep sets." Advances in neural information processing systems 30 (2017).

[2] Ilse, Maximilian, Jakub Tomczak, and Max Welling. "Attention-based deep multiple instance learning." International conference on machine learning. PMLR, 2018.

[3] Chen, Xiwen, et al. "TimeMIL: Advancing multivariate time series classification via a time-aware multiple instance learning." ICML'24


- The empirical section mainly compares against early Transformer-style and S4/LongConv models. Given the rapid development of efficient long-sequence and cross-sectional architectures, incorporating or at least discussing stronger, contemporaneous methods would be encouraged.

**Questions:**

- Did you compare early-vs-late insertion or stacking multiple Set layers? Any diminishing returns?
- Will synthetic generator, equity feature pipelines, and mortgage preprocessing be released to enable reproduction?
- This is just an open question: can the Set-Sequence concept be verified beyond financial or factor-style datasets?
- To be honest, I am not sure about the assumption of exchangeability in real-world applications. Are they only approximately exchangeable or any exceptions?

---

> ### Author Response · Authors · 2025-11-27
> **Rebuttal by Authors**
>
> We thank reviewer VTYC for their insightful review of our paper, and answer their questions below.
>
> #### W1: The Set-Sequence architecture extends the idea of multiple instance learning mean pooling to the temporal domain. There are a few related works that are encouraged to be discussed or compared.
>
> TLDR: [1] In MIL, a set has one global label; in our setting, each set element has one label. [2] We also need to exploit a time axis that is shared across our set elements (not part of the MIL setup).
> In more detail, on line 130 in our paper we compare with the Deep Sets paper. We are using similar set modules, but they are operating over time series, which is not mentioned at all in the Deep Sets paper. In addition, the output of the set module is augmented to the original features of each unit, which also does not appear in the Deep Sets paper.
> Multiple Instance Learning ([2], [3]) deals with a bag of instances for which a single label is assigned. In our setting, each bag of time series does not have a label associated but rather we produce a set summary that is augmented to the individual features of a multivariate time series model.
>
> [1] Zaheer, Manzil, et al. "Deep sets." Advances in neural information processing systems 30 (2017).
> [2] Ilse, Maximilian, Jakub Tomczak, and Max Welling. "Attention-based deep multiple instance learning." International conference on machine learning. PMLR, 2018.
> [3] Chen, Xiwen, et al. "TimeMIL: Advancing multivariate time series classification via a time-aware multiple instance learning." ICML'24
>
> #### W2: The empirical section mainly compares against early Transformer-style and S4/LongConv models. Given the rapid development of efficient long-sequence and cross-sectional architectures, incorporating or at least discussing stronger, contemporaneous methods would be encouraged.
>
> TLDR: Any sequence model can be plugged into the seq component of SetSeq. These benefits should translate immediately.
> In detail,  in our experiments, a main focus is that any multi-variate time series/sequence model can be augmented by the set module, allowing it to improve the models performance. We think that by showing this improvement (see table 1) for a varied set of backbones (convolution, SSM, hybrid, Transformer), it’s clear that the gains of using the set module works for a varied range of models. We further run an experiment with Mamba for each unit in the same equities setting as in our paper, the results are as follows, showing significant performance improvement over the Mamba model.
>
> | Model               | Sharpe Ratio | Return % | Std Dev % | Beta   | Daily Turnover | Short Fraction |
> |---------------------|--------------|----------|-----------|--------|-----------------|----------------|
> | mamba               | 2.63         | 9.00     | 3.42      | 0.05   | 1.23            | 0.47           |
> | Set-Sequence (Ours) | 4.82         | 13.0     | 2.69      | 0.028  | 0.91            | 0.48           |
>
> **Table 4:** Comparison of mamba and Set-Sequence. All training parameters were identical across models.
>
> ####Q1: Did you compare early-vs-late insertion or stacking multiple Set layers? Any diminishing returns?
> In our experiments, we use 6 set-sequence layers. Each layer first does the set module and then the sequence module. We will clarify this in the submission. In the common response we provide an ablation on using a the set module in a subset of the layers, finding the performance robust.
>
> #### Q2: Will synthetic generator, equity feature pipelines, and mortgage preprocessing be released to enable reproduction?
> Yes, all the code will be released, it’s also currently available in the supplementary material if you want to reproduce the experiments in the paper.
>
> #### Q3: This is just an open question: can the Set-Sequence concept be verified beyond financial or factor-style datasets?
> Yes, we believe that the sequence model can work well in a range of other domains with a similar data structure of a set of multivariate units that have joint dependencies.
> One domain could be predicting user churn, where each user is represented by a feature vector evolving in time. Here, each user is a unit, and we predict the churn of the user both based on the characteristics of the user (for instance, the user may be using the product less over time), as well as a population level set summary (that may for instance capture for instance that there is a general tendency to churn since the last product update that was not received well).
>
> #### Q4: To be honest, I am not sure about the assumption of exchangeability in real-world applications. Are they only approximately exchangeable or any exceptions?
> Thank you for bringing this up. As several other reviewers had similar questions we provide a common response to all of them at the top of the rebuttal.

---

> > ### Comment · Reviewer_VTYC · 2025-11-27
> >
> > Thanks for the response. The authors addressed my concerns, and I believe this paper is valuable, so I will keep my score.

---

### Official Review · Reviewer_b938 · 2025-10-31

**Soundness:** 2
**Presentation:** 2
**Contribution:** 2
**Rating:** 4
**Confidence:** 4

**Summary:**

The paper introduces a Set-Sequence model that specifically learns the cross-sectional structure in time series. The model breaks down the task into two prongs: summarizing the unit set and modeling each unit’s dynamics conditioned on their features and learned summaries. It is scalable to variable numbers of units. Empirical results show that the model outperforms the alternatives and provides interpretable summaries.

**Strengths:**

- The proposed model addresses the challenge of cross-sectional modeling.

- Linear scaling in the number of units is a practical advantage.

- The expressivity result under exchangeability maps naturally to factor-model intuition: cross-sectional moments summarize latent common variation.

**Weaknesses:**

- The assumption of exchangeability is strong, which may not generally hold in real markets across all periods.

- It’s unclear how stable the model is across market regimes, such as bull vs. bear and calm vs. crisis. Performance may be different at distinct periods and events.

**Questions:**

- Does the permutation-invariant property hold across common financial time series and data in other domains?

- How does the set summary learned by the model relate to known economic structures?

- What is the model’s performance across market regimes?

- Do conclusions hold using different window sizes and set-summary sizes?

---

> ### Author Response · Authors · 2025-11-27
> **Rebuttal by Authors**
>
> Thank you for your review and thoughtful suggestions. We answer your questions below.
>
> #### W1. The assumption of exchangeability is strong, which may not generally hold in real markets across all periods.
> Thank you for bringing this up, as several other reviewers raised similar concerns we have addressed them with a common response in the top of the rebuttal.
>
> #### W2. It’s unclear how stable the model is across market regimes, such as bull vs. bear and calm vs. crisis. Performance may be different at distinct periods and events.
> For our equities case study, in Figure 8 in the appendix, we show the cumulative returns over the full sample period from 2002 to 2021. In the plot, we see that the Set-Sequence model has robust performance in both bear markets (for instance 2007 - 2009), and bull markets. We see that adding the set module gives better risk adjusted return compared with just using the LongConv model.
> For the mortgage risk prediction task, we do a robustness analysis in Figure 15 in the appendix. The figure shows that while the average AUC varies between 0.64 and 0.71 over the 20 year period, the performance improvement compared with the baselines is stable at around 4 points per year.
>
> #### Q1: Does the permutation-invariant property hold across common financial time series and data in other domains?
> Thank you for bringing this up, as several other reviewers asked similar questions, we will address this in a common response at the top of the rebuttal.
>
> #### Q2: How does the set summary learned by the model relate to known economic structures?
> Thank you for the question. One example of an economic structure captured by the Set-Sequence model is shown in Figure 12 in the appendix of the paper. In the Figure, which is on the mortgage risk prediction task, we show that the set variable has a 70% correlation with the lagged foreclosure rate over the dataset, a known source of cross asset relationships. This indicates that the Set Summary learns the lagged foreclosure rate across loans as this is an important indicator.
>
> #### Q3: What is the model’s performance across market regimes?
> For the mortgage risk prediction task, we perform an extensive per-year performance analysis in Figure 15 of the appendix. The figure shows that while there is a variation in the AUC over years, for all years, the gain over the baseline models is consistently strong.
>
> #### Q4: Do conclusions hold using different window sizes and set-summary sizes?
> Thank you for raising this point, we provide additional experiments showing the results are robust, the results and a detailed answer is in the common response, as another reviewer also had similar questions.

---

### Author Response · Authors · 2025-11-27
**Common Response by Authors**

We would like to thank all the reviewers for their time and effort, which have meaningfully contributed to making our paper stronger. Below we address questions shared between multiple reviewers.

### Validity of the Exchangeability

*Reviewer comments*: "The assumption of exchangeability is strong, which may not generally hold..."; "Does the permutation-invariant property hold across common financial time series?"; "I am not sure about the assumption of exchangeability in real-world applications. Are they only approximately exchangeable or any exceptions?"; "How does the exchangeability assumption affect performance when dependencies are localized or non-exchangeable?"

*Author response*: Please note that *exchangeability* is a way of expressing the following statement in mathematical terms: `our problem instance is of a set of units (say, stocks) with each unit represented by some features (a time series of feature vectors)`. The primary point here is "set", i.e. the order in which the units appear does not matter. For example, if we want to predict the evolution of mortgages {A,B} based on their history, their order does not matter and thus our model output must be equivariant in the input order. So the model output would be $Y=[Y_A,Y_B]$ for input $X=[X_A, X_B]$, and it would be permuted to $Y'=[Y_B, Y_A]$ for input $X'=[X_B, X_A]$.

We note that exchangeability in our setup holds across the units (not over time) and by itself does not involve any assumptions. Given the confusion our terminology has caused to several reviewers, perhaps avoiding that term (and replacing it with a descriptive sentence like above) is a good idea? We look forward to your suggestions.


### Hyperparameter Sensitivity

*Reviewer comments*: "Do conclusions hold using different window sizes and set-summary sizes?"; "Sensitivity to look-back parameter 𝐿: The paper fixes 𝐿=3 without justification. How sensitive is the model to this choice? Does increasing 𝐿 enhance long-term dependency modeling or simply increase computation? A short sensitivity analysis would clarify its role."

*Author response*: Below we provide three sensitivity analyses, first on the look-back parameter L, with the dimension of the set-summary fixed, then with the look-back parameter fixed and ablating over the set summary dimension, and finally on the number of Set-Sequence layers that contain the Set module. We conduct the ablation on our synthetic mortgage risk task. We find that the results are rather robust to a range of choices of both the look-back parameter and the set summary dimension. We will add these additional results to the paper.
| set module lookback (L) | KL(true\|predicted) |
|------------------------:|---------------------|
| 1                       |            0.00020         |
| 2                       |             0.00023        |
| 3                       |              0.00018       |
| 5                       |              0.00042       |
| 10                      |             0.00026       |
| 20                      |            0.00027        |
| 50                      |            0.00029         |
**Table 1: Set module look-back ablation (set summary dimension = 2)**

| set summary dimension | KL(true\|predicted) |
|----------------------:|---------------------|
| 2                     |            0.00018         |
| 3                     |           0.00022          |
| 5                     |            0.00021        |
| 10                    |           0.00019          |
| 20                    |           0.00048          |
| 50                    |        0.00019           |
**Table 2: Set summary dimension ablation (set module look-back = 3)**

Finally, we provide an ablation on the number of layers where we add the set-module. We use 6 sequence layers, and a subset of the layers also have a set module.

| # Set-Module Layers | KL(true\|predicted) |
|--------------------:|--------------------:|
| 0 | 0.0013 |
| 1 | 0.00026 |
| 2 | 0.00068 |
| 3 | 0.00028 |
| 4 | 0.00037 |
| 5 | 0.00021 |
| 6 | 0.00018 |
**Table 3: KL(true‖predicted) across number of set-module layers**

---

### Meta-Review · Area_Chair_fJ92 · 2026-01-09

**Summary:**

The reviewers, while none of them been excited about the paper acknowledged the computational efficiency of the proposed approach to model cross-sectional dependencies in multi-time series. However, they also raised important concerns about i) the clarity and practicality of the exchangeability assumption; and iii) the limited empirical evaluation. During the rebuttal, the authors clarified point i), and added an ablation study on the sensitivity to hyperparameters. However, I believe that the misisng comparison with related work, as pointed out by reviewer eEp8, is a major and still open concern that the authors have disregarded with not  very convincing arguments. Since the technical and theoretical novelty is rather limited, I believe that a proper empirical evaluation and comparison with related works is the minimum for this paper to be above the acceptance bar.

**Reviewer Concerns:**

As mentioned above, I believe that the very limited empirical comparison of the proposed method remains as a major open concern.

**Reviewer Scores:**

None of the reviewers was particularly excited about the paper, nor engaged during the rebuttal. So I do not have any grounds to believe that they would have raised their scored, specially given the limited improvements on the paper performed during the rebuttal period.

---

### Decision · Program_Chairs · 2026-01-26

Reject